# Swiss halocarbon emissions for 2019 to 2020 assessed from regional atmospheric observations

Dominique Rust[1,2], Ioannis Katharopoulos[1,3], Martin K. Vollmer[1], Stephan Henne[1], Simon O'Doherty[4], Daniel Say[4], Lukas Emmenegger[1], Renato Zenobi[2], Stefan Reimann[1]

[1]Laboratory for Air Pollution/Environmental Technology, Empa, Swiss Federal Laboratories for Materials Science and Technologies, Dübendorf, Switzerland
[2]Department of Chemistry and Applied Biosciences, ETH, Swiss Federal Institute of Technology, Zurich, Switzerland
[3]Institute for Atmospheric and Climate Science, ETH, Swiss Federal Institute of Technology, Zurich, Switzerland
[4]Atmospheric Chemistry Research Group, University of Bristol, Bristol, UK

*Correspondence to*: Stefan Reimann (stefan.reimann@empa.ch)

**Abstract.** Halocarbons contribute to global warming and stratospheric ozone-depletion. They are emitted to the atmosphere by various anthropogenic activities. To determine Swiss national halocarbon emissions, we applied top-down methods, which rely on atmospheric concentration observations sensitive to the targeted emissions. We
present 12 months (September 2019 to August 2020) of continuous atmospheric observations of 28 halocarbons from a measurement campaign at the Beromünster tall tower in Switzerland. The site is sensitive to the Swiss Plateau, which is the most densely populated area of Switzerland. Therefore, the measurements are well-suited to derive Swiss halocarbon emissions. Emissions were calculated by two different top-down methods, a tracer-ratio method (TRM) with carbon monoxide (CO) as the independent tracer, and a Bayesian inversion (BI), based on
atmospheric transport simulations using FLEXPART–COSMO. The results were compared to previously reported top-down emission estimates, based on measurements at the high-Alpine site Jungfraujoch, and to the bottom-up Swiss national greenhouse gas (GHG) inventory, as annually reported to the United Nations Framework Convention on Climate Change (UNFCCC). We observed moderately elevated concentrations of chlorofluorocarbons (CFCs) and hydrochlorofluorocarbons (HCFCs), both banned from production and consumption in Europe. The
corresponding emissions are likely related to the ongoing outgassing from older foams and refrigerators and confirm the widespread historical use of these substances. For the major hydrofluorocarbons (HFCs) HFC-125 ($CHF_2CF_3$) and HFC-32 ($CH_2F_2$), our calculated emissions of $100 \pm 34$ Mg yr$^{-1}$ and $45 \pm 14$ Mg yr$^{-1}$ are in good agreement with the numbers reported in the Swiss inventory, whereas for HFC-134a ($CH_2FCF_3$) our result of $280 \pm 89$ Mg yr$^{-1}$ is more than 30 % lower than the Swiss inventory. For HFC-152a ($CH_3CHF_2$), our top-down
result of $21 \pm 5$ Mg yr$^{-1}$ is significantly higher than the number reported in the Swiss inventory. For the other investigated HFCs, perfluorocarbons (PFCs), $SF_6$ and $NF_3$, Swiss emissions were small and in agreement with the inventory. Finally, we present the first country-based emission estimates for three recently phased-in, unregulated hydrofluoroolefins (HFOs), HFO-1234yf ($CF_3CF=CH_2$), HFO-1234ze(E) ((E)-$CF_3CH=CHF$) and HCFO-1233zd(E) ((E)-$CF_3CH=CHCl$). For these three HFOs, we calculated Swiss emissions of $15 \pm 4$ Mg yr$^{-1}$, $34 \pm 14$
Mg yr$^{-1}$, and $7 \pm 1$ Mg yr$^{-1}$, respectively.

## 1   Introduction

Anthropogenic halocarbons are emitted to the atmosphere through a wide range of industrial and consumption-based activities. They are commonly used as cooling agents in refrigeration and air conditioning, as foam blowing

agents, in fire extinguishers or as solvents (WMO, 2018). Halocarbons with long atmospheric lifetimes have considerable global warming potentials (GWPs). In addition, chlorine or bromine-containing halocarbons, e.g. chlorofluorocarbons (CFCs), hydrochlorofluorocarbons (HCFCs), and brominated alkanes (halons), act as strong ozone-depleting substances. The production and consumption of these substances is regulated under the Montreal Protocol (MP) on Substances that Deplete the Ozone Layer (UNEP, 2020). Hydrofluorocarbons (HFCs), which are not ozone depleting but strong greenhouse gases (GHGs), are part of the Kigali Amendment to the MP and of the Kyoto Protocol (KP) under the framework of the United Nations Framework Convention on Climate Change (UN-FCCC). Whereas the Kigali Amendment foresees a binding down-scaling of global production and consumption of HFCs, the KP only prescribes annual reporting as emission inventories and reduction of emissions, in a basket together with other GHGs. The submitted national emission inventories are so-called "bottom-up" derived accountings of the individual halocarbon emissions, based on production, sales and consumption statistics, and emission factors (Weiss et al., 2011; Bergamaschi et al., 2018). Because of the growing pressure to reduce HFC emissions, a new generation of unsaturated HFCs has been marketed for about a decade – the hydrofluoroolefins (HFOs). These substances have short lifetimes and low GWPs and, therefore, are not regulated. In EU member states, however, they are subject to reporting under the F-gas regulation (EU, 2014).

Among the most important CFCs and halons, CFC-11 ($CCl_3F$) and CFC-12 ($CCl_2F_2$) were extensively used globally as foam blowing and cooling agents, respectively, while the minor CFC-13 ($CClF_3$) and CFC-115 ($CClF_2CF_3$) were only applied for special refrigeration purposes (Vollmer et al., 2018). H-1211 ($CBrClF_2$) and H-2402 ($CBrF_2CBrF_2$) were used as fire extinguishing agents. Regarding the HCFCs, HCFC-22 ($CHF_2Cl$) was mostly used as a cooling agent in domestic and commercial refrigeration. HCFC-141b ($CH_3CCl_2F$) and HCFC-142b ($CH_3CClF_2$) were used as foam blowing agents, and HCFC-141b was additionally applied as a solvent in different cleaning applications. HCFC-124 was, besides others, used as cooling agent in special applications. Whereas the CFCs and halons were phased-out in developed countries from 1995 onwards, the HCFCs are now also banned for use in new equipment in Europe, with an allowance of 0.5 % of the 1989 base level consumption until 2030 for maintenance of existing refrigeration and air conditioning systems. However, CFCs, halons, and HCFCs remain in many long-lived products, from which they are still continuously emitted (WMO, 2018; Montzka et al., 2021).

Among the HFCs, HFC-134a ($CH_2FCF_3$) is applied as a cooling agent in (auto)mobile air conditioners, where it has gradually replaced CFC-12 since the early 1990s. In addition, it is used as an aerosol propellant in metered-dose inhalers and as a foam blowing agent (Simmonds et al., 2017; WMO, 2018; Li et al., 2019). HFC-125 ($CHF_2CF_3$) is widely applied in refrigerant blends for stationary air conditioners, and HFC-32 ($CH_2F_2$), mainly together with HFC-125, is a component of refrigerant blends that were introduced as substitute for HCFC-22 in stationary air conditioning (Graziosi et al., 2017). HFC-134a, HFC-125, and HFC-32 are among the most widely used refrigerants in Europe. HFC-152a ($CH_3CHF_2$) is mainly used as a foam blowing agent, but also as an aerosol propellant and in refrigeration blends, replacing CFCs and HCFCs (Simmonds et al., 2016). Compared to the other HFCs, HFC-152a has the lowest lifetime of 1.6 years (Supplement 1). HFC-245fa ($CHF_2CH_2CF_3$) and HFC-365mfc ($CH_3CF_2CH_2CF_3$) are mainly used as foam blowing agents, substituting the phased-out HCFC-141b and CFC-11 (Vollmer et al., 2006, 2011; Stemmler et al., 2007). HFC-227ea ($CF_3CHFCF_3$) and HFC-236fa ($CF_3CH_2CF_3$) are mainly used as fire extinguishing agents as replacement for the halons (Laube et al., 2010; Vollmer et al., 2011), with smaller usage as propellants or special cooling agents. HFC-4310mee is used as a cleaning agent for various applications, replacing CFC-113 ($CCl_2FCClF_2$), HCFC-141b, and methyl chloroform (Le Bris et al., 2018), and as a solvent for specific uses, replacing PFCs (Arnold et al., 2014). Of the PFCs, PFC-

($C_2F_6$) and PFC-318 (c-$C_4F_8$) are emitted from the semiconductor industry, while the former is also emitted from aluminum production. The production of aluminum is also the main anthropogenic source of PFC-14 ($CF_4$). $NF_3$ is emitted from the electronics industry, and $SF_6$ is applied in electrical insulation and additionally emitted from the magnesium and aluminum industries. The unregulated HFO-1234yf ($CF_3CF=CH_2$), HFO-1234ze(E) ((E)-$CF_3CH=CHF$), and HCFO-1233zd(E) ((E)-$CF_3CH=CHCl$) were first detected in the atmosphere by Vollmer et al. (2015), using measurements from Dübendorf and Jungfraujoch (Switzerland). While HFO-1234yf is currently applied as a refrigerant in mobile air conditioners to replace HFC-134a, and in refrigerant blends (together with HFC-134a, HFC-125, and HFC-32), HFO-1234ze(E) is mainly used in refrigerant blends for residential and commercial air conditioners (together with HFC-32, HFC-152a, and isobutane), and as a foam blowing agent and propellant. Although HCFO-1233zd(E) contains a chlorine atom, its contribution to stratospheric ozone depletion is minimal due to its short atmospheric lifetime (Supplement 1).

To monitor the long-term, large-scale trends of halocarbons in the atmosphere, the global measurement network Advanced Global Atmospheric Gases Experiment (AGAGE), consisting of 15 remote background sites, and a measurement program operated by the National Oceanic and Atmospheric Administration (NOAA), have been established (Prinn et al., 2000, 2018; NOAA, 2021). Based on these atmospheric observations, the effectiveness of the regulative measures are monitored by means of the changing trends of the global atmospheric concentrations of the halocarbons.

To assess bottom-up derived halocarbon emissions on the global or the hemispheric scale with atmospheric observations, box and inverse modelling methods were developed (WMO, 2010; Bergamaschi et al., 2018; Prinn et al., 2018). This "top-down" quantification complements the existing bottom-up national and global emission inventories and adds to the credibility of reported emission inventories, or can be used to cover incomplete reporting. To derive a more robust confirmation of emissions, efforts to narrow down the top-down modelling methods to subcontinental, regional or country scale have been made (e.g. Weiss et al., 2011; Brunner et al., 2017; Bergamaschi et al., 2018). However, an important limitation in the development of top-down methods for subregional scale is the sparsity of regional observations. To improve national emission estimates, high-frequency in situ measurements are needed at higher spatial resolution, which regularly capture pollution events from the region of interest. To determine European continental emissions, measurements from the AGAGE stations Mace Head (Ireland), Tacolneston (England), Jungfraujoch (Switzerland), Monte Cimone (Italy), and Ny-Ålesund (Spitsbergen) have previously been used for top-down modelling (e.g. Lunt et al., 2015; Brunner et al., 2017; Graziosi et al., 2017; Manning et al., 2021). In addition, temporary, regional measurement campaigns have been conducted to further constrain emissions from Eastern Europe (Keller et al., 2012) and the Eastern Mediterranean (Schoenenberger et al., 2018). Furthermore, continuous in situ measurements of halogenated trace gases started at the Taunus Observatory in central Germany in 2018 (Lefrancois et al., 2021) with the aim of improving observational sensitivity of large parts of Germany, the Benelux region and France (Schuck et al., 2018). In the United Kingdom (UK) and Ireland, the Deriving Emissions linked to Climate Change (DECC) network, of which Mace Head and Tacolneston are also part, was launched in 2012. The continuous measurements of GHGs at these sites are used for top-down estimations of UK emissions (e.g. Manning et al., 2021). However, the number of (European) countries for which emissions are specifically quantified based on regional observations, is still very limited, and in Europe, UNFCCC reported inventories are complemented with top-down estimates only for the UK and Switzerland (Bergamaschi et al., 2018).

In Switzerland, until now, top-down halocarbon emission estimates were calculated based on continuous measurements at the high-altitude research station at Jungfraujoch (Reimann et al., 2021). However, Jungfraujoch is located at a topographically complex location that is only periodically influenced by direct air transports from the polluted Swiss boundary layer. Hence, emission estimates for Switzerland are based on only a few transport events reaching Jungfraujoch, mostly during the summer months (Reimann et al., 2004). In addition to halocarbons, Swiss emis-

sions of methane ($CH_4$) and nitrous oxide ($N_2O$) have been estimated annually by Bayesian inverse modelling since 2013 and 2017, respectively (Henne et al. 2016). These estimates include observations from the tall tower site Beromünster and are reported as part of the Swiss National Inventory Report (NIR) to UNFCCC.

In this study, we present continuous halocarbon measurements at the tall tower of Beromünster, in a rural area in central Switzerland (Berhanu et al., 2016). From September 2019 to August 2020, hourly measurements of 59

halocarbons, which are monitored within the AGAGE network, were performed. Air samples were analysed with a Medusa pre-concentration unit coupled to gas chromatography (GC) and mass spectrometry (MS; Miller et al., 2008; Arnold et al., 2012). We present the atmospheric records of four CFCs, two halons, four HCFCs, 10 HFCs, three perfluorocarbons (PFCs), $SF_6$, $NF_3$, and three HFOs. Based on the measurements, Swiss emissions for these 28 halocarbons were calculated by two different top-down approaches: a tracer-ratio method (TRM), with carbon

monoxide (CO) as the tracer, and a Bayesian inversion (BI). Where possible, the emissions were compared to the Swiss bottom-up inventory, reported to the UNFCCC and to the top-down estimates from Jungfraujoch (Reimann et al., 2021). Simulated maps generated from the BI modelling indicated the spatial distribution of Swiss emissions for selected halocarbons of different generations and applications.

## 2   Materials and methods

### 2.1   Measurement site

The measurements of atmospheric halocarbons were performed at the Blosenberg tall tower, close to Beromünster, Switzerland, at a sample inlet height of 212 m above ground level (a.g.l.). The tower is located in a rural area in the middle of the Swiss Plateau at 47.2° N, 8.2° E, at an altitude of 797 m a.s.l. The site was originally equipped with GHG and meteorological measurements at five different heights up to 212 m a.g.l. as part of the CarboCount-

CH project (Oney et al., 2015; Berhanu et al., 2016). Later, it was integrated into the Swiss National Air Pollution Monitoring Network (NABEL), the European Monitoring and Evaluation Program (EMEP), and the Global Atmosphere Watch (GAW) regional station network of the World Meteorological Organisation (WMO). The area referred to as the Swiss Plateau is a landscape north (N) of the Swiss Alps, south (S) of the Jura Mountains and confined by Lake Geneva in the southwest (SW) and Lake Constance in the northeast (NE). Covering one third of

the Swiss surface area, the Swiss Plateau represents the most densely populated (more than two-thirds of the Swiss population) and the industrially most active region of Switzerland. Medium and large size industrial enterprises are located in this area. Cities with the highest population on the Swiss Plateau are Zürich (population: 415 000), about 40 km NE, Lucerne (population: 82 000) about 20 km south-southeast (SSE), and Bern (population: 144 000), about 70 km SW.

Wind observations collected at 212 m a.g.l. during the measurement campaign are summarized in Fig. 1. During the campaign, the major wind direction at Beromünster was from west-southwest (WSW), followed by wind from the direction of NE, with maximum wind velocities of 24 m s$^{-1}$ at the top of the tower (212 m). This bimodal wind

direction distribution is typical for the Swiss Plateau with the near surface flow being channeled by the mountain ridges to the south (Alps) and north (Jura).

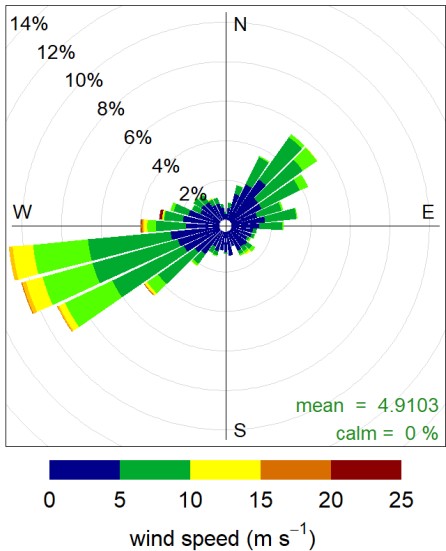

**Frequency of counts by wind direction (%)**


**Figure 1: Wind rose for the Beromünster station for the duration of the measurement campaign and based on 10 min average wind observations. The frequencies of wind from different directions are given as percentage, wind speeds are given in colored intervals in m s⁻¹.**

Beromünster was determined to be sensitive to the major part of the Swiss Plateau (Fig. 2 and Oney et al., 2015)

and is thus well-suited to capture halocarbon emissions from this area on a small to medium regional scale.

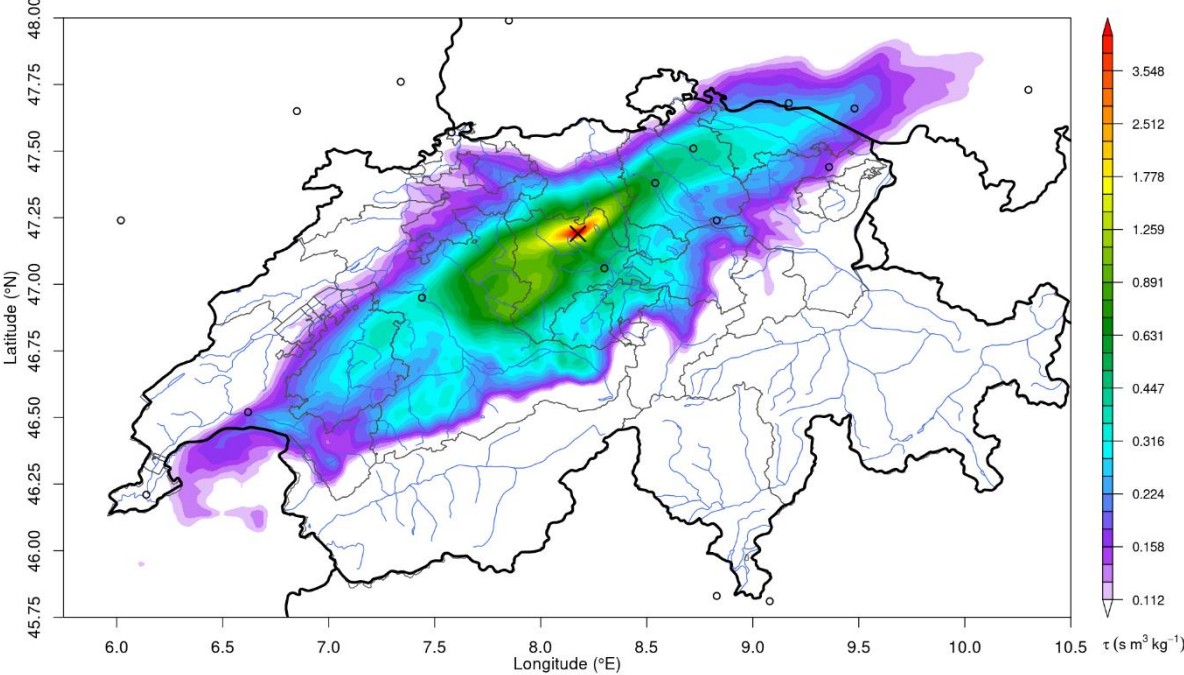

**Figure 2: Modelled total surface sensitivity for Beromünster (black cross) for the duration of the measurement campaign as obtained from the FLEXPART simulation (Sect. 2.3). The surface sensitivity is given as particle residence time per air density and surface area.**

Additional halocarbon measurements used for the inverse modelling in this study (Sect. 2.3 and Sect. 2.5) came

from the AGAGE measurement stations at Jungfraujoch (46.5º N, 8.0º E, 3580 m a.s.l., inlet height –15 m a.g.l.,

i.e. inlet height is below instrument height), Mace Head (53.3° N, –9.9° E, 8 m a.s.l., inlet height 10 m a.g.l.), and Tacolneston (52.5° N, 1.1° E, 56 m a.s.l. inlet height 185 m a.g.l.) (Prinn et al., 2018).

## 2.2     Sampling and analysis

In situ halocarbon measurements were conducted with a Medusa pre-concentration unit, coupled to GC and MS (Miller et al., 2008; Arnold et al., 2012; Prinn et al., 2018). Ambient air was sampled at 212 m a.g.l. through a continuously flushed tube, consisting of an innermost ethylene copolymer coating on a slit overlapping aluminum layer and an outer high-density polyethylene jacket (SERTOflex, 12 mm OD, 8 mm ID). From this sample stream, 2 L of air were diverted to the Medusa every 70 min at a flow rate of 100 mL/min. To prevent condensed water

from entering the sampling module, the main air stream was drawn through a custom-built in-line water trap consisting of a stainless-steel dip tube, which was regularly checked for liquid water. The integrity of the sampling line was confirmed by comparing the in situ measurements with those of simultaneously drawn flask samples from a second sampling line at the same inlet height. Additionally, all joints which connected the sampling line to the measurement devices, were sprayed with high-concentration gaseous tracers and found leak-free.

In the AGAGE measurement setup, the air sample is pre-concentrated at low temperatures in a two-trap system (Medusa). Our Medusa used a Stirling cooler (Ametek Sunpower, CryoTel-GT), leading to sample trapping temperatures of –165 °C for the first, and –180 °C for the second trap. After pre-concentration, the analytes were flushed into a GC (Agilent 6890N), using helium as the carrier gas (He 6.0, Messer Switzerland), and detected by quadrupole electron ionization MS (Agilent 5975) in selected ion mode (SIM). Chromatographic separation was

achieved with a CP-PoraBOND Q column (0.32 mm i.d. x 25 m, 5 µm film thickness, Agilent). GCWerks was used as instrument control and data processing software.

To correct for short-term instrumental drift, the measurements of two consecutive ambient air samples were bracketed by measurements of a working standard. The working standard consisted of ambient air compressed into 34 L internally electro-polished and humidified stainless steel tanks (Essex Industries Inc., USA), using an oil-free air

compressor (SA-6, RIX Industries, USA). These real-air working standards are collected at Rigi–Seebodenalp, Switzerland (47.1° N, 8.5° E, 1030 m a.s.l) during relatively clean-air conditions. To better track the MS sensitivities for low-abundant substances like HFO-1234yf, HFO-1234ze(E), and HCFO-1233zd(E), we spiked these substances into the standards. For HFO-1234yf, HFO-1234ze(E), and HCFO-1233zd(E) the respective final working standard concentrations were on average 1.5, 5.0, and 0.6 ppt. The working standards used at Beromünster were

cross-calibrated within the AGAGE relative calibration scale by a predefined inter-calibration routine, which consists of a hierarchy of calibration standards provided by the Scripps Institution of Oceanography (SIO) (Miller et al., 2008). In addition, some substances were calibrated on primary scales produced by the Swiss Federal Laboratories for Materials Science and Technologies (Empa), the Federal Institute of Metrology in Switzerland (METAS) or the University of Bristol (UB).

Three types of uncertainties were considered: the accuracy of the primary calibration scale originating from the production of the primary standard, the uncertainty from the propagation of the calibration standards within the AGAGE calibration hierarchy (Prinn et al., 2018), and the measurement precisions. The uncertainties are listed in Supplement 2. If the accuracy of the primary calibration scale was unknown for a substance, it was set to 2 %. If the uncertainty of the standard propagation was unknown, the measurement precision was propagated instead.

Measurement precisions were derived from the difference of the pair of working standard measurements that were

bracketing the actual air measurements. They were below 3 % for 26 of the 28 reported substances and below 5 % for HFC-4310mee and HFC-236fa.

The analytical setup was operated in a temperature-controlled (± 2 °C) trailer adjacent to the tower. R410a (a mixture of HFC-32 ($CH_2F_2$) and HFC-125 ($CHF_2CF_3$), 50 % by weight each) was used as refrigerant for the air conditioner. Weekly trailer indoor air measurements suggested the absence of any major refrigerant leaks.

For the use as tracer in the tracer-ratio method, CO was measured as part of the NABEL network with a Picarro G5310 analyzer (Picarro Inc., USA) using cavity ring-down spectroscopy (CRDS). Daily instrument calibration was performed with three working standards of different concentrations. These working standards were produced at Empa and calibrated against reference standards of the GAW Programme.

## 2.3    Atmospheric transport simulations

Receptor-oriented backward simulations with the Lagrangian Particle Dispersion Model (LPDM) FLEXPART (Stohl et al., 2005; Pisso et al., 2019) were carried out to estimate the sensitivity of the observed atmospheric concentrations to regional emissions (source sensitivities, Seibert et al., 2004). Depending on the location of the sites, different versions of the model, driven by different meteorological input fields, were utilized. For the Swiss sites in complex terrain (Jungfraujoch and Beromünster), the FLEXPART version adjusted for input from the regional numerical weather prediction (NWP) model COSMO was used with meteorological analysis fields operationally provided hourly by the Swiss national weather service (MeteoSwiss) at a horizontal resolution of approximately 7 km by 7 km (COSMO-7). For the two additional sites located on the British Isles (Mace Head and Tacolneston), which is towards the northwestern domain boundary of the COSMO-7 domain, the main version of FLEXPART (version 9.2_Empa) for use with input from the European Centre for Medium-range Weather Forecasts (ECMWF) Integrated Forecasting System (IFS) was applied. The employed input fields had a horizontal resolution of 0.2° by 0.2° in the Alpine area (4° W to 16° E and 39° N to 51° N) and 1° by 1° elsewhere. For both model versions, a similar backward simulation strategy was followed, releasing 50 000 model particles during 3-hourly intervals at each measurement location and tracing these particles backward in time for 8 and 10 days for FLEXPART–COSMO and FLEXPART–IFS, respectively.

The resulting source sensitivities, $m_{i,j}$, connect the spatial distribution of the emissions ($E_{i,j}$) with the concentration of a tracer in the receptor ($\chi$) via a linear relationship

$$\chi = \sum_{i,j} m_{i,j} E_{i,j} + \chi_b, \tag{1}$$

where $\chi_b$ is the average concentration of the particles at the end points of the backward simulation and is comparable to boundary conditions in Eulerian model simulations. Here, $\chi_b$ was not explicitly simulated or taken from a larger scale model, but is replaced by an observation-based baseline concentration (Sect. 2.5).

Besides total receptor concentrations, spatially resolved FLEXPART source sensitivities were used to identify situations in which air masses sampled at Beromünster were dominated by surface contact over the Swiss domain. For this purpose, the sum of $m_{i,j}$ for different land regions was calculated for each model interval and the fraction of Swiss residence time to total residence time over land was determined.

## 2.4    Emission estimation by the tracer-ratio method (TRM)

The TRM relates emissions of a target analyte to those of a tracer with known emissions during pollution events. The underlying assumption of the TRM is that both the target analyte and the tracer have similar spatial and tem-

poral emission sources and are transported to the receptor, i.e. the measurement site, without any significant additional production or loss (Yao et al., 2012). This also implies that the analyte and the tracer behave similarly in the atmosphere or that the transport distance to the measurement site is either short enough for the analyte and tracer ratio to be preserved or long enough so that analyte and tracer emissions from multiple sources are well-mixed. In this case, a sufficiently large catchment area is needed for substances with distinct emission areas, to result in improved mixing with the tracer. Overall we consider Beromünster as an adequate site for the TRM because the Swiss pollution sources are at a significant distance (i.e. Lucerne as the nearest large town 20 km away) and thus well mixed, and because the observed substances are very stable in the atmosphere and thus not modified during their transport.

In this study, CO was used as the reference tracer. It was assumed that in Switzerland CO is only emitted anthropogenically during the combustion of biofuels and fossil fuels in road transport and non-industrial combustion processes, leading to relatively constant CO emissions for all seasons (Guevara et al., 2021). Other sources of CO, i.e. emissions from wildfires or oxidation of methane and non-methane volatile organic carbons (VOCs), were assumed negligible or constant for the Swiss domain (Oney et al., 2017). The main sink of CO is oxidation with hydroxyl radicals, resulting in an atmospheric lifetime of 22 d in summer in the northern hemisphere (Miller et al., 2012), which is much longer than the transport time of CO from Swiss emission sources to Beromünster. Hence, it was possible to neglect CO degradation in the current approach.

The emissions ($E_x$) of a target analyte in units of Mg yr$^{-1}$ were calculated by applying Eq. (2):

$$E_x = E_{CO} \frac{\Delta X}{\Delta CO} \frac{M_x}{M_{CO}}. \tag{2}$$

$\Delta X$ (in units of ppt = pmol/mol) and $\Delta CO$ (in units of ppb = nmol/mol) are the respective enhancements of atmospheric concentrations above background of the target analyte and CO, respectively. To determine enhancement above background, the CO data were first averaged within the sampling time interval of each halocarbon measurement. Pollution events were then distinguished from background concentrations by applying the statistical "robust extraction of baseline signal" (REBS) method (Ruckstuhl et al., 2012). Since this method yields the atmospheric background level as a "baseline", the two expressions are used synonymously in the following. To determine a method uncertainty, different baseline variations were calculated and incorporated into the emission estimation. REBS settings included the use of a tuning factor (b = 3.5) and three different settings of the temporal window width, i.e. a bandwidth parameter of 15, 30, and 60 days. Next to a smooth baseline, the REBS method provides a global estimate of the uncertainty of the baseline. To add to the determination of the method uncertainty, different subsets of data above baseline were created by scaling the REBS uncertainty to different levels by multiplying with specific factors, i.e. 1, 1.5, and 2. The fractions of measured baseline concentrations relative to the total number of data points and the average background concentrations for the considered halocarbons for a specific example of REBS settings are given in Supplement 4. Multiplication with the ratio of molecular weights ($M_x$ and $M_{CO}$; both in units of g mol$^{-1}$) is needed to convert from mass to volume mixing ratios. $E_{CO}$ is the a priori emission inventory value of CO. To cover the timeframe of the measurement campaign, the CO inventory values for the years 2019 and 2020 were weighted accordingly, to result in a Swiss CO emission value of 152.9 Gg yr$^{-1}$. Correspondent to Reimann et al. (2021), the yet unreported Swiss CO inventory value for 2020 was calculated from the latest available CO inventory values reported to CLRTAP/EMEP and the average trend over the preceding three years.

For the emission calculation, the data were filtered based on simulated near-surface residence time of the sampled air masses (Sect. 2.3). Near-surface residence times were evaluated by country/region. Residence times over Switzerland were divided by the total residence time over land areas in the model domain. Two different minimal ratios of Swiss residence time were used for observation filtering, i.e. for events with 50 % and 75 % relative country residence times, respectively. A map, depicting the distribution of the relative country residence times of the air masses measured at Beromünster for different regions, is given in Supplement 4. The air measured at Beromünster resided most of the time over Switzerland, France and Germany.

To calculate the halocarbon–CO emission ratio all remaining pollution events above baseline were summed up to give a term for $\Delta X$ and $\Delta CO$, respectively, and the two terms were then divided. The values arising from the different baseline and country residence time settings were averaged to give a final emission value.

Several sources of error were taken into account throughout the tracer-ratio calculations. For the halocarbon and CO measurements, the corresponding measurement precisions (Sect. 2.2) at 1-sigma (68 %) confidence level, and the uncertainty of the modelled baseline fit were propagated by standard Gaussian error propagation. Then the two types of calibration accuracies (Sect. 2.2) for the halocarbon measurements were added to the uncertainty of the term $\Delta X$ before calculation of the halocarbon–CO emission ratio. Final uncertainties for emission estimations were calculated at the 2-sigma (95 %) confidence level. As described before, method uncertainties arising from the choice of parameters for baseline fitting and relative country residence time were incorporated by taking into account the different settings for the baseline fitting parameters and the two different levels of relative country residence times.

For CFC-13, CFC-115, H-2402, HCFC-22, HCFC-124, HFC-23, HFC-236fa, PFC-318, PFC-14, and NF$_3$, only a very limited number of pollution events with significant magnitude were observed. Thus, depending on the set REBS and country residence time parameters, the number of data points included in the emission estimation with the TRM was greatly reduced (Supplement 4). Emissions calculated with less than 10 data points were deemed unreliable.

### 2.5    Emission estimation by Bayesian inverse (BI) modelling

The second top-down approach for estimating Swiss emissions utilizes the FLEXPART simulated source sensitivities coupled with a Bayesian inversion (BI) framework. The latter is used to obtain an optimized state of the emissions combining a priori knowledge of the emissions, model simulated source sensitivities, and observations at the receptor sites. The method applied in this study was extensively described by Henne et al. (2016) for methane and frequently applied to halocarbon emissions (e.g., Park 2021, Simmonds 2020, Rigby 2019, Schönenberger 2018).

The inversions in this study cover the period of the field campaign in Beromünster. Daily mean values of the observations at Beromünster, Jungfraujoch, Tacolneston, and Mace Head were utilized. Daily mean observations were preferred over the use of short aggregation intervals (e.g., 3-hourly) because little changes in total and spatially resolved emissions were seen when using the latter. The use of the longer aggregates reduces the inverse problem size and, hence allows for a faster and, in our experience, more robust estimation of covariance parameters. Moreover, sensitivity inversions for HFO-1234yf were performed in order to quantify the sensitivity of the a posteriori emissions in Switzerland to the selection of measurement sites. When adding additional observations from the Taunus Observatory in central Germany or when removing the observations from the British Isles, changes in total Swiss emissions were smaller than 5 %.

Modelled source sensitivities for the four receptor sites were used together with an a priori estimate for the emissions $x_b$ and the observations $\chi_o$ in a BI framework, in order to obtain an optimized state for the emissions. The inversion domain extends from 12.0º W to 21.1º E and 36.0º N to 57.5º N. The a posteriori emissions were calculated by minimizing the following cost function

$$J = \frac{1}{2}(x - x_b)^T B^{-1}(x - x_b) + \frac{1}{2}(Mx - \chi_o)^T R^{-1}(Mx - \chi_o),\tag{3}$$

where $x$ denotes the a posteriori state vector comprised of gridded emissions and baseline concentrations, $x_b$ gives the a priori state, M corresponds to the source sensitivities, $\chi_o$ to the observations, and B and R are the uncertainty covariance matrices of the a priori emissions and the combined model-observation uncertainty, respectively. Both B and R are symmetric block matrices. Matrix B contains two blocks, $B^E$ and $B^B$, describing the uncertainty covariance of the gridded emissions and the baseline, respectively. The diagonal elements of matrix $B^E$ are proportional to the a priori emissions, whereas the off-diagonal elements are assumed to be spatially correlated with an exponential decay with the distance,

$$B_{i,i}^E = \left(f_E x_{b,i}\right)^2\tag{4}$$

$$B_{i,j}^E = e^{-\frac{d_{i,j}}{L}}\sqrt{B_{i,i}^E}\sqrt{B_{j,j}^E}, \qquad i \neq j.\tag{5}$$

Factor $f_E$ is optimized during a maximum likelihood step (Henne et al., 2016). Factor $d_{i,j}$ is the distance between grid cells and L is the correlation length optimized again in the maximum likelihood step. The diagonal elements of block $B^B$ are set to a fixed value proportional to an estimate of the baseline uncertainty, and the non-diagonal elements are assumed to be correlated with an exponentially decreasing correlation of the baseline uncertainty between baseline nodes at a given site,

$$B_{i,i}^B = f_B \sigma_b^2\tag{6}$$

$$B_{i,j}^B = e^{-\frac{T_{i,j}}{\tau_b}}\sqrt{B_{i,i}^B}\sqrt{B_{j,j}^B}, \qquad i \neq j,\tag{7}$$

where $T_{i,j}$ is the time difference between the nodes and $\tau_b$ is the temporal correlation length. The factors $f_B$ and $\tau_b$ are optimized during the maximum likelihood step.

Block matrix R contains as many block matrices as the number of the receptor sites (in our case four), and each block matrix contains both model and observations uncertainty covariances. Diagonal elements of R contain both observations and model uncertainties, while off-diagonal elements are again assumed to be correlated with an exponentially decreasing structure with time,

$$R_{i,i} = \sigma_0^2 + \sigma_{min}^2 + \sigma_{srr}^2 \chi_{p,i}^2\tag{8}$$

$$R_{i,j} = e^{-\frac{T_{i,j}}{\tau_0}}\sqrt{R_{i,i}}\sqrt{R_{j,j}}, \qquad i \neq j,\tag{9}$$

where $\sigma_0$ is the uncertainty of the observations, while $\sigma_{min}$ and $\sigma_{srr}$ are uncertainties related to the transport model optimized again during the maximum likelihood step. Factor $T_{i,j}$ is the time difference between measurements and $\tau_0$ is the temporal correlation length set to 0.01 days, reflecting very low autocorrelation when using daily average observations. No covariance between different observing sites was considered.

The minimization of the cost function Eq. (3) reduces the difference between observed and simulated values $(\chi_o, Mx)$, additionally constrained by the difference between the a priori and a posteriori emission estimates. A posteriori emissions are given by

$$x = x_b + BM^T(MBM^T + R)^{-1}(\chi_0 - Mx_b).\tag{10}$$

For this study, the a priori baseline of each halocarbon was determined by the REBS method (Sect. 2.4) and was optimized according to the corresponding site. REBS settings included the use of the tuning factor b = 3.5, a temporal window width of 30 days, and a maximum of 10 iterations. For Beromünster, the baselines calculated for Jungfraujoch were applied, because synoptic scale baselines are needed to follow particles to their endpoints of the simulation outside the inversion domain. The a priori emissions used for the HFCs (except HFC-152a), PFCs,

$SF_6$, and $NF_3$ were taken from the 2018 national reports to the UNFCCC (UNFCCC, 2021) for the reporting countries in the model domain. Since for HFC-134a a major difference between the 2018 and 2019 inventories is reported, an additional test inversion was run using the 2019 inventory values. However, this had a negligible effect on the result and a priori values for Switzerland were applied as listed in Table 1. For HFC-152a, the Swiss inventory value was not used as a priori since for this substance emissions are attributed to the manufacturing and not the emitting countries. For HFC-152a, CFCs (except CFC-13), halons, HCFCs, and HFOs, the 2019 Swiss emis-

sions estimated based on the Jungfraujoch data (Reimann et al., 2021), and as annually submitted to the Swiss Office for the Environment (FOEN), were used as a priori. For CFC-13 the 2020 value was used as a priori as the 2019 value indicated depletion. The a priori emissions of these substances for the remaining countries in the domain were distributed according to the Swiss emissions on a per-capita basis. A priori uncertainties were obtained

through the above-mentioned maximum likelihood approach. Because the Jungfraujoch-based estimates rely on the TRM that is based on a very limited number of data points (Reimann et al., 2021), sensitivity inversions were implemented to assess the variability of the emissions if the confidence interval of the Jungfraujoch-based a priori emission values was large. For the sensitivity testing, half and double of the a priori estimates for all countries in the inversion domain were used. The sensitivity simulations showed only little response to the used a priori value,

therefore only the results with the mean a priori simulations are reported here. Except for HFO-1234yf, the HFOs were treated as inert for the inversions, assuming that the transport times from emission sources to BRM are sufficiently small to avoid larger chemical losses. Monthly average atmospheric lifetimes of HFO-1234yf as based on Henne et al. (2012) were used to update the source sensitivities specifically for this compound. Subsequently, these updated source sensitivities were used in the inversion. Resulting Swiss emissions were about 10 % higher

than when assuming inert HFO-1234yf. The other HFOs treated here have longer atmospheric lifetimes (Supplement 1) and, hence, their lifetime impact on Swiss emissions is smaller and was deemed negligible in the light of other uncertainties.

    For all the halocarbons in this study, population-based a priori emissions were utilized. Flat a priori distributions (homogeneous within each country and zero over the ocean) were tested for the subset of HFC-23, $SF_6$, and PFC-

14. However, the simulation results were inferior in terms of performance, most likely due to the specific topographic situation in Switzerland, where population distribution and industrial activity are closely linked and separated from the high altitude regions of the Alps. Thus, the population-based distribution was pursued. The inversion output provided gridded emissions for the major part of the European domain. To calculate the Swiss emissions, the emissions from the grids lying inside the Swiss borders were extracted.

The inversions performance and reliability of results for the different substances was assessed through several statistical measures. In Supplement 5 the most important statistical measures and the inversions reliability are summarized. The $\chi^2$ index (defined as, $\chi^2 = J(x)^2/_d$, $d$ being the number of observations) assesses the probability density distribution of the a posteriori model residuals and a posteriori emission differences, which should follow a $\chi^2$ distribution with mean equal to $d/2$ (e.g. Berchet et al., 2013). Ideally, a $\chi^2$-index with a value close to one

can be achieved with well-chosen covariance matrices. The degree of freedom (DF) describes the reduction in the

normalized error of the a priori values due to the number of available observations, and thus, is a measure of improvement of the a posteriori when compared to the a priori. Two additional comparison statistics for the a posteriori simulated versus observed regional (above baseline) concentrations at Beromünster were considered: the correlation coefficient, $r^2$, and the normalized standard deviation, nSD, which is the standard deviation of the simulations divided by the standard deviation of the observations. For both statistics, values close to one indicate favorable model performance. Based on the abovementioned parameters, the reliability of the inversion calculations for the different substances was evaluated. Inversion results with $r^2$ greater than 0.1 were deemed reliable, which was the case for most of the substances. Exceptions were CFC-13, CFC-115, H-2402, HCFC-124, HFC-23, HFC-236fa, PFC-318, and PFC-14, for which reported results should not be over-interpreted.

**Table 1: For each halocarbon, the following information is listed: the measurement sites included in the Bayesian inverse (BI) modelling, i.e. Beromünster (BRM), Jungfraujoch (JFJ), Tacolneston (TAC) and Mace Head (MHD), and the source of the a priori value, i.e. either taken from the Swiss inventory as reported to the UNFCCC (UNFCCC, 2021), or taken from the results calculated within the CLIMGAS project, based on the Jungfraujoch measurements (Reimann et al., 2021) and (Reimann, 2021). The respectively used a priori value for Switzerland is rounded to two significant figures.**

| substance | sites included in inversion | a priori information | a priori value for Switzerland ($Mg\ yr^{-1}$) |
|---|---|---|---|
| CFC-11 | BRM, JFJ, TAC, MHD | CLIMGAS | 50 |
| CFC-12 | BRM, JFJ, TAC, MHD | CLIMGAS | 21 |
| CFC-13 | BRM, JFJ, TAC, MHD | CLIMGAS | 0.70 |
| CFC-115 | BRM, JFJ, TAC, MHD | CLIMGAS | 1.0 |
| H-1211 | BRM, JFJ, TAC, MHD | CLIMGAS | 2.0 |
| H-2402 | BRM, JFJ, TAC, MHD | CLIMGAS | 0.30 |
| HCFC-22 | BRM, JFJ, TAC, MHD | CLIMGAS | 37 |
| HCFC-141b | BRM, JFJ, TAC | CLIMGAS | 6.0 |
| HCFC-142b | BRM, TAC, MHD | CLIMGAS | 25 |
| HCFC-124 | BRM, JFJ, TAC, MHD | CLIMGAS | 0.20 |
| HFC-134a | BRM, JFJ, TAC, MHD | UNFCCC | 510 |
| HFC-125 | BRM, JFJ, TAC, MHD | UNFCCC | 130 |
| HFC-32 | BRM, JFJ, TAC, MHD | UNFCCC | 51 |
| HFC-152a | BRM, JFJ, TAC, MHD | UNFCCC | 18 |
| HFC-245fa | BRM, JFJ, TAC, MHD | UNFCCC | 0.20 |
| HFC-365mfc | BRM, JFJ, TAC, MHD | UNFCCC | 4.7 |
| HFC-23 | BRM, JFJ, TAC, MHD | UNFCCC | 0.60 |
| HFC-227ea | BRM, JFJ, TAC, MHD | UNFCCC | 1.9 |
| HFC-236fa | BRM, JFJ, TAC, MHD | UNFCCC | 0.50 |
| HFC-4310mee | BRM, JFJ, TAC, MHD | UNFCCC | 0.40 |
| PFC-116 | BRM, JFJ, TAC, MHD | UNFCCC | 0.50 |
| PFC-318 | BRM, JFJ, MHD | UNFCCC | 0.020 |
| PFC-14 | BRM, JFJ, TAC, MHD | UNFCCC | 0.60 |
| $SF_6$ | BRM, JFJ, TAC, MHD | UNFCCC | 7.0 |
| $NF_3$ | BRM, JFJ, MHD | UNFCCC | 0.050 |
| HFO-1234yf | BRM, JFJ, TAC, MHD | CLIMGAS | 8.0 |
| HFO-1234ze(E) | BRM, JFJ, TAC, MHD | CLIMGAS | 12 |
| HCFO-1233zd(E) | BRM, JFJ, TAC, MHD | CLIMGAS | 3.0 |

 ## 3    Results and discussion

### 3.1    Measured time series

The one-year Beromünster records are shown in Fig. 3 and Supplement 3 for the halocarbons with the highest atmospheric abundance and/or for which emissions were calculated. This includes four CFCs, two halons, four HCFCs, 10 HFCs, three PFCs, $SF_6$, $NF_3$, three HFOs, and CO. Concentrations are reported as dry air mole fractions in pmol mol$^{-1}$ (ppt) for the halocarbons, and in nmol mol$^{-1}$ (ppb) for CO.

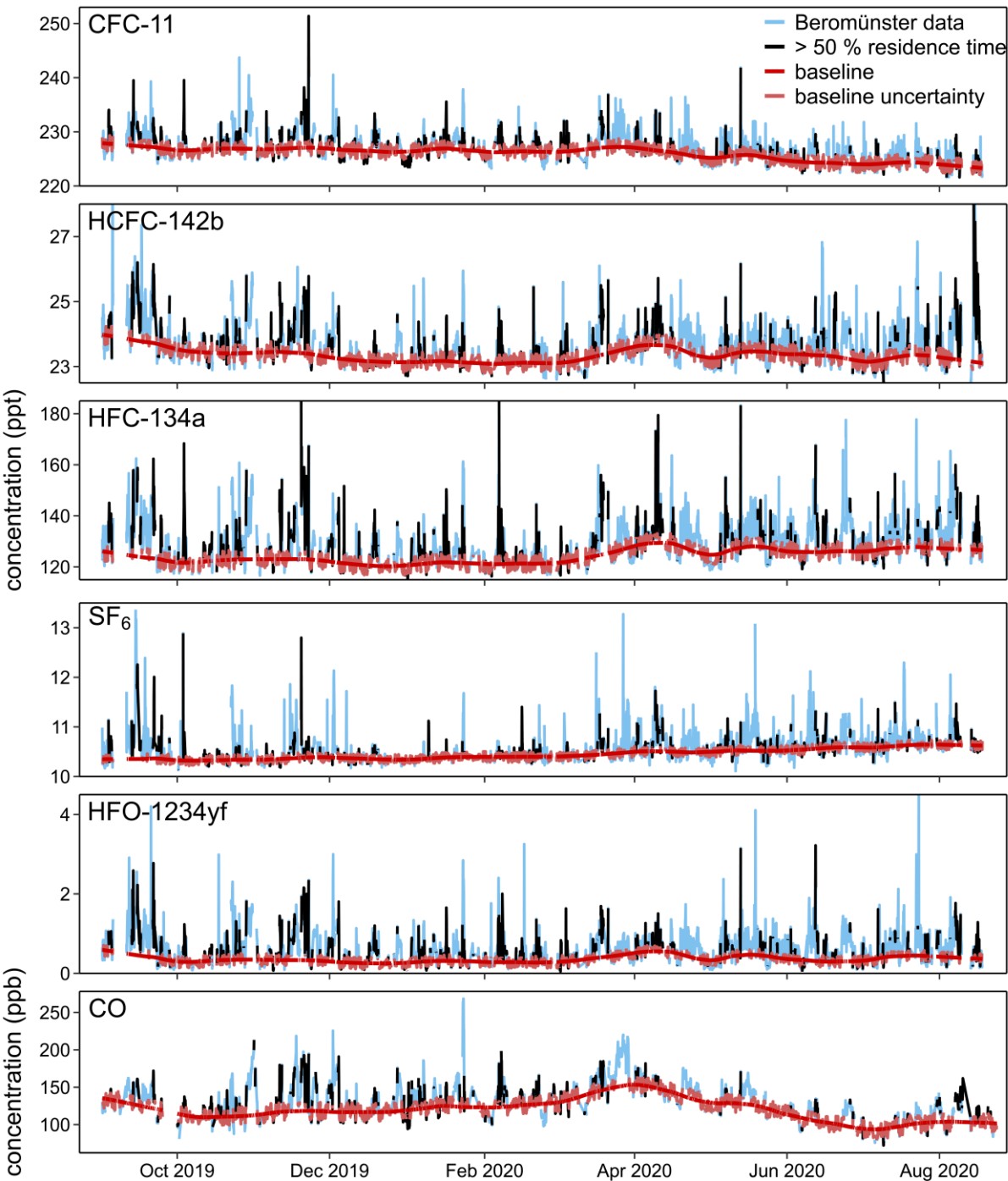

**Figure 3: One-year time series of the atmospheric concentrations measured at Beromünster. Air samples with more than 50 % relative residence time in Switzerland are highlighted in black. The baselines (red lines) calculated with a REBS bandwidth parameter of 30 days are shown with their uncertainty bands (light red) derived with the REBS multiplication factor set to 1.5.**

The majority of the records show pollution events during the whole measurement campaign. For HCFC-142b, HFC-134a, HFC-125, HFC-32, HFC-152a, HFO-1234yf, HFO-1234ze(E), and HCFO-1233zd(E), the pollution events quickly succeed each other within hourly or daily time intervals. Therefore, for these substances, a background concentration level is scarcely reached during the whole year, or only seasonally during the winter months.

This can be explained by the extensive use of these halocarbons in various applications and their continuous emissions from regional sources captured by the site. In the following, the individual substances are discussed in more detail.

For CFC-11, CFC-12, and H-1211, we were able to define solid baselines with some overlying structure arising from ongoing emissions of existing installations (banks), underlining their extensive historical use. For CFC-13,

CFC-115, and H-2402, pollution events were virtually absent, indicating minor historic use.

In addition, HCFC-22, HCFC-141b, and the minor HCFC-124 ($CHClFCF_3$) showed few distinctive pollution events, whereas for HCFC-142b pollution events were more frequent with enhancements above baseline reaching a maximum larger than 6 ppt. Our observations underline the finding that these HCFCs are still emitted from outgassing of existing foams and refrigeration units.

For the HFCs, our measurements show the highest abundances and most frequent pollution events for HFC-134a, HFC-125, and HFC-32, with average background concentrations of around 120, 40, and 30 ppt, respectively. HFC-134a reached the highest pollution events of more than 80 ppt over baseline, whereas pollution events of HFC-125 and HFC-32 were smaller with maximum values of approximately 20 and 30 ppt. For HFC-152a we observed an average baseline concentration of about 10 ppt with only minor enhancements. This can be explained

by small HFC-152a emissions in Switzerland arising only from outgassing of existing foams, as there is no production of foams and the use as propellant is forbidden. HFC-245fa, HFC-365mfc, HFC-227ea, HFC-236fa, and HFC-4310mee all showed clear baselines with a few distinctive pollution events, which, however, only lasted for a few hours. For these five HFCs, the majority of the emissions were localized outside of Switzerland – only 19 % of the pollution events had a Swiss residence time above 50 %, and 5 % above 75 % residence time (REBS band-

width of 30 days and multiplication factor of 1.5). For HFC-227ea, we observed a few distinctive pollution events with a Swiss residence time larger than 50 %. Although the specified events were exceeding the baseline by about 1 ppt only, they may point at non-continuous, recurrent emissions of HFC-227ea. As one of the more abundant HFCs in the atmosphere, HFC-23 ($CHF_3$) showed an average baseline level of approximately 34 ppt. However, there were only very few pollution events. This can be ascribed to the absence of HCFC-22 synthesis in Switzer-

land, from which HFC-23, besides minor emissions from specific other applications (Oram et al., 1998; Miller et al., 2010; Montzka et al., 2010; Stanley et al., 2020), is mainly emitted as an unwanted by-product (Keller et al., 2011; WMO, 2018).

For $SF_6$, sporadic pollution episodes were observed, with only 17 % of the pollution events greater than 1 ppt, however, showing a high contribution from Switzerland. For $NF_3$ and PFC-14, the time series show little variability

beyond measurement precision. PFC-14 is emitted during the production of aluminum, while $NF_3$ is produced during manufacturing processes in the electronic industry, both absent from within the footprint of the station. The other investigated PFCs are used in the semiconductor industry: PFC-116 shows sporadic pollution events, especially during the summer months and PFC-318 only shows two discernible events in March and July 2020. For both substances only about 20 % of the pollution events are attributed to Switzerland with more than 50 % country

residence time.

For the newly produced HFO-1234yf, HFO-1234ze(E), and HCFO-1233zd(E), the average baseline levels were as low as 0.4, 0.5, and 0.2 ppt, respectively. More than 50 % of the measurements were assigned to pollution events. The highest pollution events were seen for HFO-1234ze(E), exceeding 40 ppt over baseline, followed by lower events of about 4 ppt for HFO-1234yf and 2 ppt for HCFO-1233zd(E). Our observations confirm the in-
creasing use of HFO-1234ze(E) and HFO-1234yf as refrigerants in Switzerland, and the fact that HCFO-1233zd(E) is only allowed for limited application as cooling agent and as foam blowing agent.

The time series of CO (used as the tracer in the TRM) shows a pronounced variability of the baseline with a maximum at the end of March and a minimum in July. Many high pollution events occur during the winter months, followed by fewer observed events during spring and summer, however not implying smaller emissions in summer
but rather reflecting the faster mixing of surface emissions throughout a larger fraction of the troposphere (Sect. 2.4). The seasonality in the baseline and in the amplitude and frequency of pollution events is typical in the northern hemisphere. The seasonal variation of CO at Beromünster was already studied in detail by Satar et al. (2016), who found smaller seasonal amplitudes compared to other European tall tower stations.

### 3.2 Emissions from Switzerland

Based on the atmospheric measurements at Beromünster, Jungfraujoch, Mace Head, and Tacolneston, Swiss emissions were determined with the TRM (Sect. 2.4) and BI models (Sect. 2.5). The results are listed in Table 2 and are compared to the Swiss top-down emission estimates based on the Jungfraujoch data (Reimann et al., 2021), and, for HFCs, PFCs, $SF_6$, and $NF_3$, to the latest (2019) bottom-up inventory reported to the UNFCCC (UNFCCC, 2021). Because the Jungfraujoch-based Swiss emissions are calculated as three-year averages, the 2019 value is
used for comparison instead of the 2020 value. The time series of the Swiss emission values since 2015 are presented in Fig. 4. For the Jungfraujoch and the inventory emissions, error bars are only depicted for the 2019 values. For the inventory values, uncertainties were assessed based on the information and uncertainties (at 1-sigma (68 %) confidence level) given in the NIR (FOEN, 2021 a) for the individual emission source categories, and on the detailed emission numbers submitted to the UNFCCC.
Figure 5 summarizes the 2019/2020 emission values, grouped for different halocarbon classes, and shows the corresponding values converted to $CO_2$ equivalents.




**Table 2: Calculated Swiss emissions for 2019/2020, including the Beromünster (BRM) measurements, with the tracer-ratio method (TRM) and Bayesian inverse (BI) modelling. The results are compared to the average of the Jungfraujoch (JFJ)-based emission values for 2019 and the Swiss inventory for 2019. The average BRM–TRM and BI values are rounded to two significant figures. Uncertainties are given at 2-sigma.**

| substance | average BRM–TRM/ BI (Mg yr$^{-1}$) 2019/2020 | BRM–TRM (Mg yr$^{-1}$) 2019/2020 | BI (Mg yr$^{-1}$) 2019/2020 | JFJ–TRM (Mg yr$^{-1}$) 2019[b] | inventory (Mg yr$^{-1}$) 2019[c] |
|---|---|---|---|---|---|
| CFC-11 | 65 ± 24 | 93 ± 18 | 38 ± 16 | 50 ± 21 | - |
| CFC-12 | 27 ± 9.5[a] | 45 ± 9 | 10 ± 3 | 21 ± 25 | - |
| CFC-13 | 1.1 ± 0.49[a] | 2 ± 0.5 | 0.02 ± 0.2 | –1 ± 2 | - |
| CFC-115 | 1.7 ± 0.75[a] | 3 ± 0.6 | 0.3 ± 0.4 | 1 ± 2 | - |
| H-1211 | 5.5 ± 2.2 | 8 ± 2 | 3 ± 1 | 2 ± 1 | - |
| H-2402 | 0.34 ± 0.12 | 0.5 ± 0.1 | 0.2 ± 0.06 | 0.2 ± 0.6 | - |
| HCFC-22 | 23 ± 9.4[a] | 40 ± 8 | 7 ± 5 | 37 ± 20 | - |
| HCFC-141b | 7.3 ± 2.8 | 12 ± 2 | 3 ± 2 | 6 ± 4 | - |
| HCFC-142b | 15 ± 5.0 | 19 ± 4 | 10 ± 3 | 25 ± 14 | - |
| HCFC-124 | 2.1 ± 0.73[a] | 4 ± 0.7 | 0.4 ± 0.2 | 0.2 ± 1 | - |
| HFC-134a | 280 ± 89 | 289 ± 58 | 274 ± 67 | 314 ± 48 | 448 ± 135 |
| HFC-125 | 100 ± 34 | 94 ± 19 | 107 ± 28 | 77 ± 11 | 124 ± 37 |
| HFC-32 | 45 ± 14 | 47 ± 9 | 44 ± 11 | 29 ± 8 | 54 ± 16 |
| HFC-152a | 21 ± 5.4 | 27 ± 5 | 15 ± 2 | 18 ± 7 | 0.4 ± 0.8 |
| HFC-245fa | 2.3 ± 1.3 | 4 ± 0.8 | 1.3 ± 1.2 | 8 ± 3 | 0.2 ± 0.1 |
| HFC-365mfc | 6.5 ± 2.8 | 8 ± 2 | 5 ± 2 | 9 ± 3 | 5 ± 12 |
| HFC-23 | 4.2 ± 2.0[a] | 8 ± 2 | 0.3 ± 0.4 | 2 ± 5 | 0.8 ± 0.2 |
| HFC-227ea | 2.0 ± 0.78 | 3 ± 0.7 | 0.9 ± 0.4 | 3 ± 1 | 1.3 ± 1 |
| HFC-236fa | 0.32 ± 0.13[a] | 0.6 ± 0.1 | 0.04 ± 0.06 | –0.1 ± 0.5 | 0.5 ± 0.2 |
| HFC-4310mee | 1.4 ± 0.57 | 2 ± 0.4 | 0.7 ± 0.4 | 0.6 ± 1.6 | 0.4 ± 0.7 |
| PFC-116 | 1.3 ± 0.57 | 2 ± 0.4 | 0.5 ± 0.4 | 0.9 ± 2.0 | 0.4 ± 0.6 |
| PFC-318 | 1.6 ± 0.58[a] | 3 ± 0.6 | 0.1 ± 0.06 | 0.8 ± 1.9 | 0.02 ± 0.04 |
| PFC-14 | 4.1 ± 1.0[a] | 8 ± 1 | 0.6 ± 0.2 | 5 ± 5 | 0.6 ± 0.9 |
| SF$_6$ | 9.8 ± 2.8 | 11 ± 2 | 9 ± 2 | 6 ± 2 | 5.4 ± 5 |
| NF$_3$ | 0.34 ± 0.16[a] | 0.6 ± 0.1 | 0.05 ± 0.06 | 0.4 ± 0.7 | 0.03 ± 0.1 |
| HFO-1234yf | 15 ± 4.2 | 14 ± 3 | 15 ± 3 | 8 ± 1 | - |
| HFO-1234ze(E) | 34 ± 14 | 42 ± 8 | 27 ± 11 | 12 ± 2 | - |
| HCFO-1233zd(E) | 7.3 ± 1.3 | 7 ± 1 | 8 ± 1 | 3 ± 2 | - |

a) Averaged BRM-TRM and BI emissions for which the uncertainty ranges do not overlap with the individual estimates.

b) (Reimann et al., 2021) and (Reimann, 2021)

c) (UNFCCC, 2021)


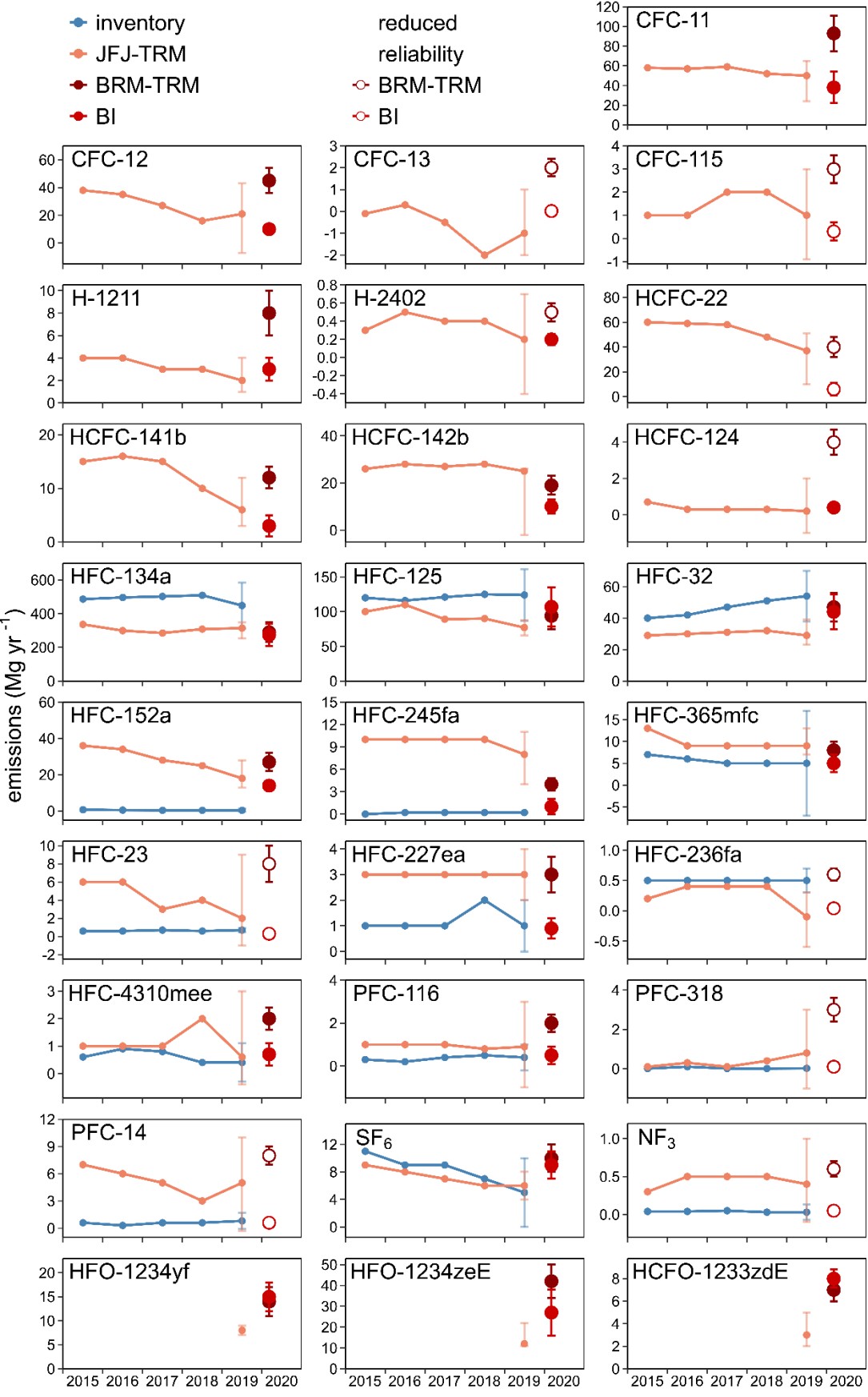

**Figure 4: Calculated Swiss emissions for 2019/2020, including the Beromünster (BRM) measurements, with the tracer-ratio method (TRM) and Bayesian inverse (BI) modelling. The results are compared to Jungfraujoch (JFJ)-based estimates and the Swiss bottom-up inventory. Results with reduced reliability are indicated.**

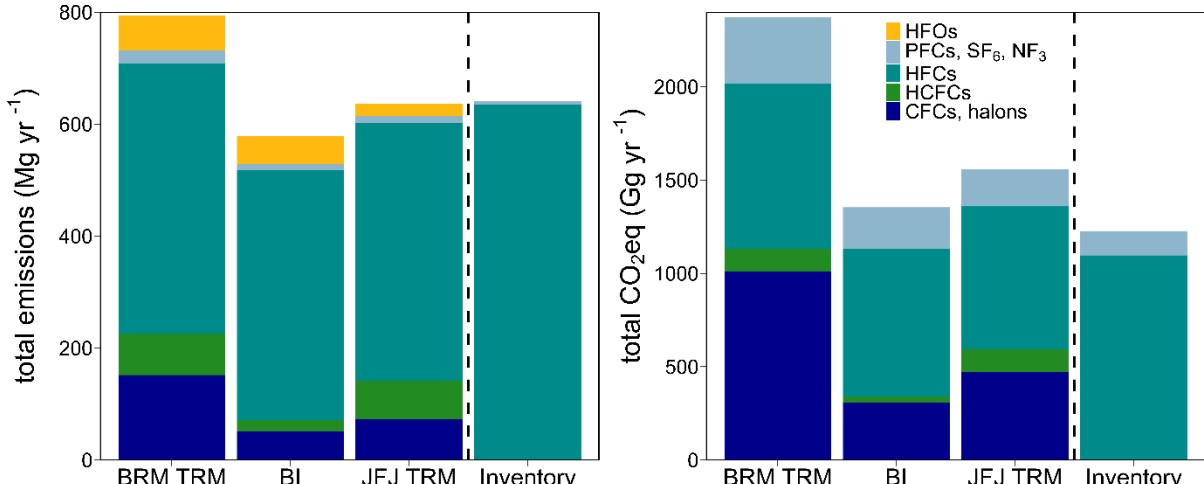


**Figure 5: The total Swiss emissions (left) and the corresponding CO₂ equivalents (right) for 2019/2020, categorized by substance classes, i.e. chlorofluorocarbons (CFCs) and halons, hydrochlorofluorocarbons (HCFCs), hydrofluorocarbons (HFCs), perfluorocarbons (PFCs), NF₃, SF₆, and hydrofluoroolefins (HFOs), for the Beromünster (BRM) and Jungfraujoch (JFJ)-based tracer-ratio method (TRM) results, the Bayesian inversion (BI) results, and the Swiss national inventory. The Swiss inventory only reports emissions for the HFCs, PFCs, NF₃, and SF₆.**


In the following, the estimated emissions are discussed in more detail. Emissions including Beromünster measurements are given as the averaged values of the TRM and BI methods. In the text, emissions are given with two significant digits.

### 3.2.1 CFCs, halons and HCFCs

With both methods, the largest emissions of this group were found for CFC-11 and CFC-12, reflecting their substantial use in the past. For both substances, the Beromünster TRM values are higher than the Jungfraujoch values, which in turn are higher than the BI results. Over the past years, the CFC-12 emissions observed at Jungfraujoch were decreasing more rapidly than those of CFC-11, which potentially is due to the more stable emissions from CFC-11 from old foams in comparison to CFC-12, which was mostly used as a refrigerant.

For CFC-13, CFC-115, H-1211, and H-2402, calculated emissions are smaller than 10 Mg yr$^{-1}$. Due to the limited number of pollution events attributed to Switzerland, the results for these four substances are highly uncertain. Furthermore, the TRM emissions for all four substances are higher than the BI and 2019 Jungfraujoch results.

For HCFC-22, HCFC-141b, HCFC-142b, and HCFC-124, the Beromünster and Jungfraujoch TRM values and the BI results diverge as well, with systematically higher TRM estimates. For HCFC-141b, for example, the TRM

value (12 Mg yr$^{-1}$), is 4 times as high as the BI value (3.0 Mg yr$^{-1}$).

### 3.2.2 HFCs, PFCs, SF₆, and NF₃

The highest emissions of the investigated HFCs were found for HFC-134a with ~280 Mg yr$^{-1}$ for both methods. This value is in good agreement with the Jungfraujoch value of 310 Mg yr$^{-1}$ (Reimann et al., 2021). The national inventory shows an increasing trend peaking at 510 Mg yr$^{-1}$ in 2018 and then drops to 450 Mg yr$^{-1}$ in 2019. Alt-

hough the 2019 imported amount of HFC-134a corresponds to the yearly import fluctuations, it has been presumed that substitute products are increasingly applied (FOEN, 2021 b). Nevertheless, the latest inventory value is still 35 % higher than the average of the three top-down values. This finding hints at an overestimation of the Swiss HFC-134a bottom-up inventory, which is further supported by other top-down modelling studies for different study domains and time periods. For the UK, Say et al. (2016) and Manning et al. (2021) found significantly lower

modelled top-down versus inventory estimates for the respective time periods of multiple years between 1995 and

2020. Upon their study results, Say et al. (2016) recommended the re-evaluation of some input-parameters of the UK inventory model. For the European domain, Graziosi et al. (2017) found lower averaged top-down emissions for HFC-134a for the years 2003 – 2014 compared with the sum of the inventories. They suggested an overestimation of emission factors used for the inventory calculations in some countries to be the reason for the high

UNFCCC reported values. Lunt et al. (2015) studied emissions on a global scale and reported 21 % lower top-down estimates for all Annex I countries as compared to the aggregated UNFCCC inventories for 2007 to 2012. They emphasized the noticeable divergence found for different countries, presumably due to differently assessed emission factors or activity data.

For HFC-125, the Beromünster TRM and the BI emission estimates are well in accordance with each other and

result in an average value of 100 Mg yr$^{-1}$. This value lies between the Jungfraujoch (77 Mg yr$^{-1}$) and inventory values (120 Mg yr$^{-1}$). Again, the measurement-based values are somewhat lower than the inventory value. Also for the UK, Manning et al. (2021) recently reported on average 35 % lower top-down than inventory values for the years 1995 to 2018.

HFC-32 shows the third highest Beromünster TRM and BI emission estimates of the investigated HFCs (45 Mg yr$^{-1}$

). This value exceeds the Jungfraujoch estimate (29 Mg yr$^{-1}$) and compares slightly better with the 2019 bottom-up inventory (54 Mg yr$^{-1}$), whereas an increasing gap was found between (lower) Jungfraujoch estimates and (higher) inventory estimates during the past years (Reimann et al., 2021), apparently due to a greater rate of growth in the inventory values.

Based on our measurements, the next largest Swiss emissions were attributed to HFC-152a. For this HFC, the

TRM and BI results differ significantly from the inventory value. The discrepancy may be due to the fact that for HFC-152a, UNFCCC reported emissions are attributed to the manufacturing, not the emitting countries, which may lead to a distortion of the calculated emissions at the country level. Similar observations of greatly differing top-down and bottom-up estimates have been published before. Manning et al. (2021) found significantly larger inventory than inverse modelling values for the UK. Investigating the global domain, Lunt et al. (2015) reported 8

times higher atmosphere-based emissions of HFC-152a for the Annex I countries for the years 2007 to 2012, but raise the objection of incomplete reporting of this substance for some of the included Annex I countries. Global and (European) regional comparisons between top-down calculated HFC-152a emissions and reported inventory values were also published by Simmonds et al. (2016) who showed varying agreement between the inverse results and reported UNFCCC values for many European countries for the years 2006 – 2014.

For HFC-245fa, HFC-365mfc, HFC-23, HFC-227ea, HFC-236fa, and HFC-4310mee, Swiss emissions were determined to be smaller than 10 Mg yr$^{-1}$. This is also the case for the Jungfraujoch and the inventory values.

Of all investigated substances, PFC-116, PFC-318, PFC-14, $SF_6$, and $NF_3$ are among those with the longest lifetime and the highest 100-year GWP. Their Swiss emissions were determined below 10 Mg yr$^{-1}$.

### 3.2.3    HFOs

As described in Sect. 3.1, HFOs are increasingly replacing HFCs in various applications (WMO, 2018). Therefore, emissions and atmospheric abundance of these gases are expected to grow in the future (Vollmer et al., 2015). Based on our measurements, we present the first emission estimates for Switzerland. They amount to 15 Mg yr$^{-1}$, 34 Mg yr$^{-1}$ and 7.3 Mg yr$^{-1}$ for HFO-1234yf, HFO-1234ze(E) and HCFO-1233zd(E), respectively. For HFO-1234yf and HCFO-1233zd(E), the 2019 and 2019/2020 Beromünster and Jungfraujoch TRM and the BI values

compare well. For HFO-1234ze(E), the Beromünster TRM result (42 Mg yr$^{-1}$) is 1.5 times as high as the BI result (27 Mg yr$^{-1}$). Both estimates are somewhat higher than the 2019 Jungfraujoch value.

### 3.2.4     Methods appraisal

Both applied methods, the TRM and the BI, have their advantages and disadvantages. For the TRM we make the assumptions that the analyte and the tracer have similar spatial and temporal emissions sources, and that the

transport distance is either sufficiently short for the ratio of analyte and tracer to be preserved until reaching the receptor, or that the transport distance is sufficiently long so that analyte and tracer emissions from multiple sources are well-mixed when reaching the receptor (Sect. 2.4). The Bayesian inversion makes the assumption that emissions are constant in time. For compounds with intermittent emissions, this may lead to reduced model performance. Furthermore, the method seeks to locate emissions in space, guided by a priori information. In the case of

large spatial differences between a priori and real emissions, the method will be challenged, once again leading to reduced model performance. We observe, that for most substances, except the major HFCs, the HFOs, and SF$_6$, the TRM result exceeds the BI result. Possible reasons for this are that the assumption of similar emission sources of analyte and CO as the tracer does not hold and/or that the analyte and tracer emissions are not well mixed when reaching the receptor, leading to a distortion of the halocarbon-tracer ratio. Nonetheless, we used CO as a good

universal tracer for many substances. If we used another dispersedly emitted tracer, we would have similar problems. For both calculation methods, we indicated the reliability of the emissions results (Sect. 2.4, 2.5, and Fig. 4) and regarding the most highly emitted substances, we especially consider our results dependable for the major HFCs, the three HFOs, and SF$_6$. With both independent top-down approaches incorporating method uncertainties due to the made assumptions, we stated the average of the individually calculated TRM and BI results as our best

emission value (Table 2), thereby also increasing the uncertainty to each emission value. In addition, we indicated where the uncertainty range of the average emission value does not overlap with the individual results, which often is the case when we pointed out reduced reliability of our calculated results.

### 3.3     Swiss source regions

A significant asset of BI is its ability to geographically locate a posteriori emission distributions. The corresponding

maps for selected halocarbons are shown in Fig. 6 (Switzerland) and in Supplement 7 (European domain). The subset of substances includes CFC-11 and HCFC-142b, as representatives for the CFCs and HCFCs, the three most highly emitted HFCs, which are used as refrigerants, the foam blowing agent HFC-365mfc, as well as SF$_6$, and three HFOs.

All a posteriori distributions continue to reflect the population-based distribution, which was used in the a priori

emissions. However, there are differences between the compounds, visible as differing pronounced hot spots. CFC-11, HCFC-142b, and HFO-1234ze(E) emissions were especially pronounced for the area around Zurich, which has the highest population density. HFC-134a, HFC-125, HFC-32, and HFO-1234yf, all used as cooling agents, were emitted more diffusely across the Swiss Plateau, showing, in addition to the more populated areas, hot spots in other regions (e.g., between Lausanne and Bern). The foam blowing agent HFC-365mfc showed some-

what less pronounced emissions from the populated areas with larger emissions from outside Switzerland dominating the distribution at a wider scale. SF$_6$ showed generally diffuse emissions from the Swiss Plateau, however with some more pronounced grids at Basel and Bern. HCFO-1233zd(E) emissions were especially emphasized in the areas of Basel, Bern, and Lausanne.

In the European frame, the Swiss emissions of CFC-11, HCFC-142b, and HFO-1234ze(E) were of comparable

magnitude as the emissions from the border areas of France and Germany, except for the significantly enhanced area around Zurich. For HFC-134a, HFC-125, HFC-32, $SF_6$, HFO-1234yf, and HCFO-1233zd(E) the Swiss emissions were of comparable magnitude as the emissions north of the border in Germany and France, with a diverse distribution of locations with increased emissions. For HFC-365mfc, compared to Switzerland, enhanced emissions were attributed to France and Germany.

The maps of the absolute differences between the a priori and the a posteriori emissions (Supplement 7) show somewhat the same population driven distribution as the relative source maps. However, for all substances except CFC-11 and HFO-1234ze(E), the model decreased the a posteriori emissions at the region around Zurich as compared to the a priori distribution. For HFO-1234ze(E), a posteriori emissions were especially increased around Zurich. For the HFCs and $SF_6$ the model increased the a posteriori emissions from the southwestern region towards

Lausanne.

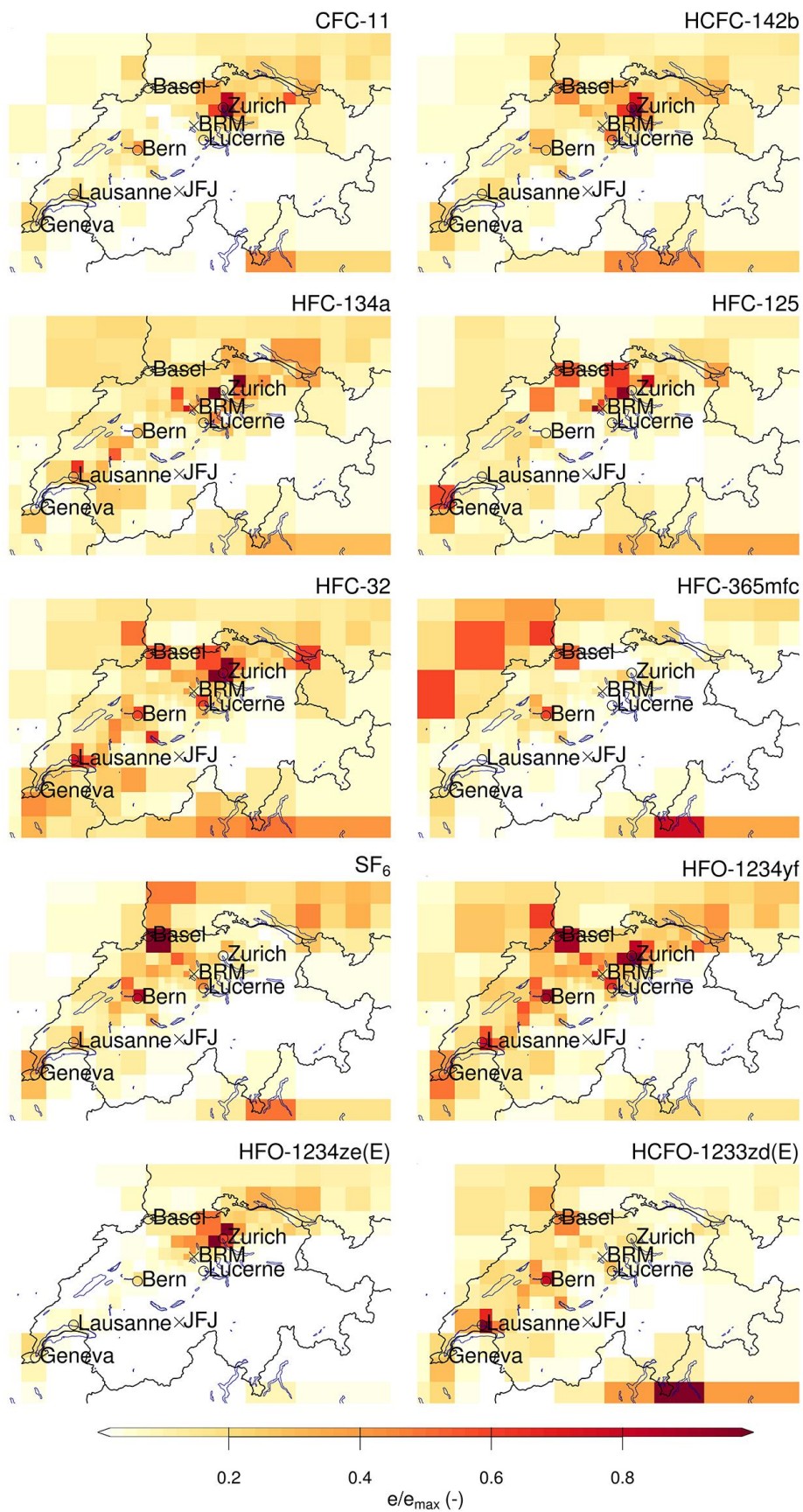

**Figure 6: Emission maps for Switzerland, generated from the Bayesian inverse (BI) modelling. Beromünster (BRM), Jungfraujoch (JFJ), and major Swiss cities are indicated. Gridded emissions (e), given in µg s⁻¹ km⁻², are scaled to the global maximum (e_max).**

## 4 Conclusions

In this study, we present the time series measured at Beromünster of four CFCs, two halons, four HCFCs, ten HFCs, three PFCs, $SF_6$, $NF_3$, and three HFOs. Based on these records, Swiss emissions were determined by two different top-down methods. The results were compared to Jungfraujoch-based estimates, and to the Swiss national inventory, as annually reported to the UNFCCC. For the major CFCs and HCFCs, our emission results are consistent with the ongoing release from remaining banks. For HFC-134a, the Beromünster TRM and the BI results compare well with the Jungfraujoch estimate, whereas they are approximately one third lower than the inventory value. For HFC-125 and HFC-32, the top-down emission results compare well to the Swiss inventory values. For HFC-152a, the top-down results were significantly larger than the Swiss inventory. For the minor HFCs, the PFCs, $SF_6$, and $NF_3$, emissions are below 10 Mg yr$^{-1}$, and, where available, compare well within the order of magnitude to the 2019 UNFCCC reported inventory values. In addition, we present the first emission estimates of the recently introduced unsaturated HFO-1234yf, HFO-1234ze(E), and HCFO-1233zd(E) for Switzerland. Total HFC emissions are in good agreement between the three top-down methods, and in varying agreement for the other substance classes. Moreover, regions with emission sources were defined for a subset of the investigated halocarbons and, for most substances, point at the more densely populated areas of Switzerland. Overall, the measurements at Beromünster provided additional information for the calculation of Swiss regional halocarbon emissions and the allocation of local source areas.

## 5 Data availability

Continuous atmospheric halocarbon measurement data for the AGAGE stations are available from (http://agage.mit.edu/data/agage-data). Measurement data for Tacolneston are available from the Centre of Environmental Data Analysis (CEDA) archive (https://catalogue.ceda.ac.uk/uuid/a18f43456c364789aac726ed365e41d1). Beromünster measurement data are available from the Zenodo data repository (https://doi.org/10.5281/zenodo.5843548).

## 6 Author contribution

DR, MKV, SOD, and DS collected and evaluated the measurement data. DR and IK estimated the Swiss halocarbon emissions by the tracer-ratio method and Bayesian inverse modeling, being intensely supported by SH and SR. DR prepared the manuscript with substantial contribution from IK and revision from LE, RZ, and all other co-authors.

## 7 Competing interests

The authors declare that they have no conflict of interest.

## 8 Acknowledgements

We thank Matthias Hill, Paul Schlauri, and Silvio Harndt from Empa for giving fundamental instrumental and technical support. We acknowledge Rüdiger Schanda from University of Bern for the constructive cooperation at

the Beromünster site. We acknowledge Henry Wöhrnschimmel and Sabine Schenker from the Swiss Federal Office for the Environment (FOEN) for valuable information regarding the Swiss halocarbon consumption and the Swiss bottom-up inventory. We thank the personnel operating the Advanced Global Atmospheric Gases Experiment (AGAGE) measurement stations at Jungfraujoch, Tacolneston and Mace Head for conducting, evaluating,

690  and providing the halocarbon measurement data. AGAGE operations are supported by the Upper Atmosphere Research Program of NASA, through grants NAG5-12669, NNX07AE89G, NNX11AF17G, and NNX16AC98G (to MIT) and NNX07AE87G, NNX07AF09G, NNX11AF15G, and NNX11AF16G (to SIO); Department for Business, Energy & Industrial Strategy (BEIS) Contract TRN 1537/06/2018 (to the University of Bristol for Mace Head and Tacolneston). Financial support for the measurements at Jungfraujoch was provided by the Swiss Na-

695  tional Programs HALCLIM and CLIMGAS-CH (FOEN) and by the International Foundation High Altitude Research Stations Jungfraujoch and Gornergrat (HFSJG). We acknowledge the Nationales Beobachtungsnetz für Luftfremdstoffe (NABEL/ FOEN, Empa) for infrastructural contributions and for providing the carbon monoxide (CO) measurement data, and MeteoSwiss for providing meteorological observations and COSMO model analysis. FLEXPART simulations were carried out at the Swiss National Supercomputing Centre (CSCS) under project

700  grants s862 and s1091. We acknowledge the Swiss National Science Foundation for funding the research for this study under the project IHALOME (SNSF, project 200020_175921).

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
