# Peer review of "Swiss halocarbon emissions for 2019 to 2020 assessed from regional atmospheric observations"

_Atmospheric Chemistry and Physics, 2021_

## Author Comment (AC1)

**Swiss halocarbon emissions for 2019 to 2020 assessed from regional atmospheric observations**

Dominique Rust[1,2], Ioannis Katharopoulos[1,3], Martin K. Vollmer[1], Stephan Henne[1], Simon O'Doherty[4], Daniel Say[4], Lukas Emmenegger[1], Renato Zenobi[2], Stefan Reimann[1]

[1]Laboratory for Air Pollution/Environmental Technology, Empa, Swiss Federal Laboratories for Materials Science and Technologies, Dübendorf, Switzerland
[2]Department of Chemistry and Applied Biosciences, ETH, Swiss Federal Institute of Technology, Zurich, Switzerland
[3]Institute for Atmospheric and Climate Science, ETH, Swiss Federal Institute of Technology, Zurich, Switzerland
[4]Atmospheric Chemistry Research Group, University of Bristol, Bristol, UK

*Correspondence to*: Stefan Reimann (stefan.reimann@empa.ch)

**Introduction**

**RC1:** "This paper describes Swiss emissions of a large number of greenhouse and ozone-depleting gases from two top-down methods and compares these results with those officially reported or derived from Jungfraujoch observations. The new observations from Beromunster, Switzerland are introduced and form the main basis for the paper, although other observation stations are used in the inverse modelling."

**Reply:** A short additional comment on the selection of the sites for the Bayesian inversion (as was also addressed in the reply to the second independent reviewer): The decision of the selection of sites was, on the one hand, based on the quality and completeness of the observations at each site and, on the other hand, on the sensitivity of the sites to Swiss emissions, the main focus of this study. We performed test inversions including an additional site (Taunus Observatory) or excluding the sites at Mace Head and Tacolneston (see also the addendum in Sect. 2.5 of the mansucript). Based on this, for the Bayesian inversion, apart from Beromünster, we continued with the sites at Jungfraujoch, Mace Head and Tacolneston, as described in the manuscript.

**General comment**

**RC1:** For quite a few gases there are very significant differences between the TRM and the BI methods. It is therefore a reasonable question to ask whether the times of good agreement are just fortuitous. The key question that has to be addressed is why there is there good agreement sometimes and poor on other occasions. This does undermine the credibility of what is presented if this is not addressed. Which method do the authors believe is the better method?

**Reply:** Thank you for addressing this substantial question. To meet this point, we added another subsection (Sect. 3.2.4 Methods appraisal) to the manuscript discussing the two calculation methods in this light: "Both applied methods, the TRM and the BI, have their advantages and disadvantages. For the TRM we make the assumptions that the analyte and the tracer have similar spatial and temporal emissions sources, and that the transport distance is either sufficiently short for the ratio of analyte and tracer to be preserved until reaching the receptor, or that the transport distance is sufficiently long so that analyte and tracer emissions from multiple sources are well-mixed when reaching the receptor (Sect. 2.4). The Bayesian inversion makes the assumption that emissions are constant in time. For compounds with intermittent emissions, this may lead to reduced model performance. Furthermore, the method seeks to locate emissions in space, guided by a priori information. In the case of large spatial differences

between a priori and real emissions, the method will be challenged, once again leading to reduced model performance. We observe, that for most substances, except the major HFCs, the HFOs, and $SF_6$, the TRM result exceeds the BI result. Possible reasons for this are that the assumption of similar emission sources of analyte and CO as the tracer does not hold and/or that the analyte and tracer emissions are not well mixed when reaching the receptor, leading to a distortion of the halocarbon-tracer ratio. Nonetheless, we used CO as a good universal tracer for many substances. If we used another dispersedly emitted tracer, we would have similar problems. For both calculation methods, we indicated the reliability of the emissions results (Sect. 2.4, 2.5, and Fig. 4) and regarding the most highly emitted substances, we especially consider our results dependable for the major HFCs, the three HFOs, and $SF_6$."

**Substantive Points**

**RC1:** P.7. L225: "for a larger emitting region and over an increased time period" – Larger than what and increased relative to what? Please can this sentence be clarified.

**New line numbering "track_changes" document:** P.8, L.263

**New line numbering "corrected" document:** P.8, L.252

**Reply:** Thank you for the remark, the phrasing was too vague. We improved the text to make it clear: " This also implies that the analyte and the tracer behave similarly in the atmosphere or that the transport distance to the measurement site is either short enough for the analyte and tracer ratio to be preserved or long enough so that analyte and tracer emissions from multiple sources are well-mixed. In this case, a sufficiently large catchment area is needed for substances with distinct emission areas, to result in improved mixing with the tracer."

**RC1:** P.8, L250: "specific sigma factors, i.e. 1, 1.5, and 2" – Are these not better described as multiplication factors? If I understand correctly this means that when the factor is 1, more points are considered 'polluted' as they are above the baseline + (baseline uncertainty)? Please can this be made clearer? In fact if table S4 (there is no Supplement 4.1) had extra columns the values for factors of 1 and 2 could be included making this very clear.

**New line numbering "track_changes" document:** P.9, L.291

**New line numbering "corrected" document:** P.8, L.279

**Reply:** In the text we changed the term "multiplying with specific sigma factors" to "multiplying with specific factors". We corrected the indication to the Supplement to "Supplement 4". In the caption for Figure 3 and on P.16, L. 499 (track changes document) we corrected the term "sigma factor" to "multiplication factor". In the Supplement, we added columns for the background fractions for the multiplication factors of 1 and 2 to Table S3, and corrected the caption text accordingly. For Figure S1 we also corrected the caption text accordingly.

**RC1:** P.8, L.272: "were added to the" – This was unclear to me, you cannot just add 2%, say, to both elements? You could add 2% to the top and subtract 2% from the bottom and vice versa, to give a range. Please can this be clarified.

**New line numbering "track_changes" document:** P.9, L.312 following.

**New line numbering "corrected" document:** P.9, L.300 following.

**Reply:** The two types of accuracies resulting from the calibration of the halocarbon measurements were added in an absolute manner to the uncertainty already being assigned to the term $\Delta X$. This latter uncertainty results from

the propagation of the measurement precisions of each halocarbon *X* and the uncertainty of the modelled baseline fit being subtracted from *X* (forming *ΔX*). We changed the wording of the text to: "For the halocarbon and CO measurements, the corresponding measurement precisions (Sect. 2.2) at 1-sigma (68 %) confidence level, and the uncertainty of the modelled baseline fit were propagated by standard Gaussian error propagation. Then the two types of calibration accuracies (Sect. 2.2) for the halocarbon measurements were added to the uncertainty of the term *ΔX* before calculation of the halocarbon–CO emission ratio. Final uncertainties for emission estimations were calculated at the 2-sigma (95 %) confidence level."

**RC1:** P.9, L.294: "0.2° by 0.2° in the Alpine area and 1° by 1° elsewhere" – How big is the Alpine area? Also 1 degree (110km) seems very coarse for modelling coastal Mace Head and Tacolneston (~50km from the coast) sites. Has this been demonstrated to be sufficient? What impact would having 0.2 degrees everywhere make?

**New line numbering "track_changes" document: :** P.8, L.243

**New line numbering "corrected" document:** P.7, L.232

**Reply:** The Alpine domain extends from 4° W to 16° E and 39° N to 51° N. The coarse resolution for Mace Head and Tacolneston was used in previous studies as well and did not reveal a strong limitation of the model to capture the observed concentrations (e.g., Simmonds et al. 2020). We added the coordinates for the Alpine area to the manuscript text accordingly.

**RC1:** P.9, L.296: "for 4 and 10 days" – How far do the particles travel in these time scales on average and as a minimum? The length of the simulation will naturally affect the residence time percentages used in the TRM? Also it seems reasonable that all the particles will have clearly left the Swiss area within 4 days but it is not clear the background will be fully mixed, i.e. a source just beyond the 4 days may still be very discernible. Can the authors be confident this is negligible?

**New line numbering "track_changes" document:** P.8, L.244

**New line numbering "corrected" document:** P.7, L.234

**Reply:** Usually, particles have left the Swiss domain after 4 days of integration. However, we agree that frequently the particles have not left the COSMO-7, central European domain after 4 days. We evaluated the contribution to residence time and concentrations at the receptor when continuing the simulation beyond day 4. When including four more days in the backward integration, we see that additional contributions to residence time and greenhouse gas concentrations ($CO_2$, $CH_4$, $N_2O$) are smaller than 10 % for the location of Beromünster. In the context of the relative residence time contribution from Switzerland employed in the tracer-ratio method, we consider this contribution negligible. Furthermore, in the context of the Bayesian inverse modelling the effect of 4 vs 8 day footprints was evaluated for a single compound (HFO-1234yf). Differences for the Swiss national emissions were again in the order of 10 %. Although, this is well within the uncertainty range given by the inverse method, we decided to update the transport simulations and inversions to include 8-day backward calculations.

**RC1:** P.9, L.315: "Daily mean values" – In previous work that I have seen using this method, 3-hourly averaging times were used. Is there a specific reason why the observations were averaged into daily values? It seems odd to lose all this extra information, I think some justification of this is necessary.

**New line numbering "track_changes" document:** P.11, L.357

**New line numbering "corrected" document:** P.9, L.319

**Reply:** Tests with 3-hourly and daily observations were performed. The differences for Swiss national total emissions were again within the a posteriori uncertainties of the inversion. Little benefits were observed in the localization of emissions when using 3-hourly vs. daily mean observations. The decision for daily data was finally based on the fact that it reduces the costs of the maximum likelihood estimate of the covariance parameters considerably and overall lead to a more robust estimation of these parameters. Furthermore, the autocorrelation of 3-hourly observations is considerably larger than that of daily averages. Using the latter allows to treat the model-data mismatch uncertainty as being uncorrelated in time. We added the following justification to the manuscript: "Daily mean observations were preferred over the use of short aggregation intervals (e.g., 3-hourly) because little changes in total and spatially resolved emissions were seen when using the latter. The use of the longer aggregates reduces the inverse problem size and, hence allows for a faster and, in our experience, more robust estimation of covariance parameters."

**RC1:** P9. L.319: "12.0o W to 21.1o E and 36.0o S to 57.5 o N" – Given that you follow the particles for only 4 days from the Swiss sites then it seems reasonable that many of the released particles will not have left your inversion domain within the 4 days. For example a direct southerly wind of 3 m/s consistently for 4 days is insufficient to move a particle from the southern edge (36deg N) to Beromunster (47degN). How is this accounted for? Also I assume it is a typo as you have written 36 degrees South, should this not be 36 degN?

**New line numbering "track_changes" document:** P.11, L.368

**New line numbering "corrected" document:** P.10, L.330

**Reply:** Thanks for spotting the typo. This was corrected accordingly. The 4 day integration time was discussed already above and has been extended to 8 days for the revised manuscript. This doubling in integration time results in an increase of simulated regional concentrations of around 10 %. For various reasons, this does not automatically translate into a 10 % decrease in Swiss emissions. First, "missing" emissions were probably attributed to more distant regions by the inversion, those were we missed sensitivity beyond day 4. Second, part of the missing regional sensitivity can be reflected by a higher baseline concentration. As the latter is adjusted through the inversion and the adjustments differ if 4 or 8 day integration times were used, the final effect of integration time on total emissions may also vary. Even with 8 day integration time there may still be cases when particles have not left the model domain completely or left the model domain on its eastern boundary, beyond which emission in Eastern Europe still contribute to the regional concentration signal observed at the sites in Western Europe. While this is a general problem of regional inversions and can be improved by sampling boundary conditions from a larger scale model, here we rely on the inverse adjustment of the concentration baselines at each site to compensate for such boundary effects.

**RC1:** P10. L351:"The HFOs were treated as inert for the inversions, assuming that the transport times from emission sources to BRM are sufficiently small to avoid larger chemical losses" – Given the very short lifetimes of some of the HFOs (single digit days in summer for HFO-1234yf), I think the authors need to quantify the potential error here and also the bias between summer and winter.

**New line numbering "track_changes" document:** P.12, L.428

**New line numbering "corrected" document:** P.11, L.385

**Reply:** When considering monthly average HFO-1234yf lifetimes (taken from Henne et al. 2021) in the calculation of the source sensitivities (footprints) Swiss national emission estimates for this compound were actually about 15

% larger than in the base case that ignores lifetimes. Since the travel times of Swiss emissions to the receptor sites Beromünster and Jungfraujoch are relatively short, the impact of lifetimes is actually limited. For regions further away from the observational sites the differences in a posteriori emissions were larger. For HFOs other than HFO-1234yf atmospheric lifetimes are considerably larger (19 days for HFO-1234ze(E) and 42.5 days for HCFO-1233zd(E)) and, hence, impact on Swiss emissions will be even smaller. Hence, for the revised manuscript we updated the inversion for HFO-1234yf including atmospheric lifetimes, but left those for the other HFOs unchanged.

We added the following comment to the manuscript: "Monthly average atmospheric lifetimes of HFO-1234yf as based on Henne et al. (2012) were used to update the source sensitivities specifically for this compound. Subsequently, these updated source sensitivities were used in the inversion. Resulting Swiss emissions were about 10 % higher than when assuming inert HFO-1234yf. The other HFOs treated here have longer atmospheric lifetimes and, hence, their lifetime impact on Swiss emissions is smaller and was deemed negligible in the light of other uncertainties."

**RC1:** P10. L354: "HFC-23, SF6, and PFC-14" – The emissions of PFC-14 from Al production and HFC-23 as an industrial bi-product will not be population based. So even though the final statistics maybe improved using a population-based prior, is the use of such a prior reasonable or in any other way justifiable? If little is assumed known about the distribution of emissions is not using a 'flat' prior more reasonable? Also I assume the authors meant 'flat, land-based' prior rather than flat across the inversion domain – please clarify.

**New line numbering "track_changes" document:** P.13, L.436

**New line numbering "corrected" document:** P.11, L.394

We carried out additional sensitivity inversions for these three substances with flat a priori distributions. However, the results were less reliable based on the parameters describing the quality of the inversion, i.e. the correlation coefficient, the chi index, the degrees of freedom, and the normalized standard deviation. In the end, flat a priori distributions are also informative and drive the solution. The choice of the a priori is very important for the estimation of the a posteriori distribution in Bayesian inversions, and if the measurements and the prior pose similar uncertainties they are more or less evenly weighted for the estimation of the posterior. Hence, even if the emissions of these substances are not exactly population distributed, the emissions will be more likely coming from the Swiss plateau and not from the Alps, making the assumption of a population based prior more justifiable. Regarding HFC-23 emissions the inversion assumes constant emissions during the year, while this is not true for this substance, making it more difficult for the inversion to handle its case. Concerning what flat means, indeed it is flat land-based. Zero emissions were assigned in the grid cells which correspond to ocean. Finally, the "flat" a priori is uniform in each different country.

Additional explanation was added to the revised manuscript (lines 437-439 in the track changes version).

**RC1:** P.18, L548: "The results compare well to the Jungfraujoch and the inventory values" – I am not convinced that this is the case for all of these gases, e.g. HFC-145fa and HFC-227ea and HFC-23. A similar comment can be made about PFC-14.

**New line numbering "track_changes" document:** P.24, L.646

**New line numbering "corrected" document:** P.19, L.586

**Reply:** Sorry, this was confusing. We meant to say that also the Jungfraujoch and the inventory values are below 10 Mg yr$^{-1}$. We adapted the text accordingly to: "For HFC-245fa, HFC-365mfc, HFC-23, HFC-227ea, HFC-236fa, and HFC-4310mee, Swiss emissions were determined to be smaller than 10 Mg yr$^{-1}$. This is also the case for the Jungfraujoch and the inventory values. Of all investigated substances, PFC-116, PFC-318, PFC-14, SF$_6$, and NF$_3$ are among those with the longest lifetime and the highest 100-year GWP. Their Swiss emissions were all determined below 10 Mg yr$^{-1}$."

**RC1:** P.19, L.560: Why is this gas so different across the methods?
**New line numbering "track_changes" document:** P.24, L.658
**New line numbering "corrected" document:** P.20, L.595
**Reply:** As described above, in the case of Beromünster, the tracer-ratio method seems to be better applicable to substances where the halocarbon and tracer ratio is influenced dispersively, i.e. with emissions from several directions and sources. However, HFO-1234ze(E) emissions are pronounced in the direction northeast of Beromünster or from Zürich. Therefore, the tracer-ratio method may not represent this HFO the best. Moreover, with the newly calculated Bayesian inversion results with 8 days integration time instead of 4 days, the tracer-ratio and Bayesian inversion results align a lot better, reducing the difference by about 50 %.

**Minor Points**

**RC1:** P.2, L.63: "it is used as aerosol propellant" – Insert "an"
**New line numbering "track_changes" document:** P.2, L.71
**New line numbering "corrected" document:** P.2, L.66
**Reply:** done

**RC1:** P.2, L.63: "and as foam blowing agent" – Insert "a"
**New line numbering "track_changes" document:** P.2, L.72
**New line numbering "corrected" document:** P.2, L.67
**Reply:** done

**RC1:** P.3, L.82: "HFO-1234yf is currently applied as refrigerant" – Insert "a"
**New line numbering "track_changes" document:** P.3, L.91
**New line numbering "corrected" document:** P.3, L.86
**Reply:** done

**RC1:** P.3,L85: "and as foam blowing agent and propellant" – Insert "a"
**New line numbering "track_changes" document:** P.3, L93
**New line numbering "corrected" document:** P.3, L88
**Reply:** done

**RC1:** P.4, L.127: "which are constantly monitored within the AGAGE network" – Remove the word "constantly", they are measured at high-frequency not constantly.

**New line numbering "track_changes" document:** P.4, L.136

**New line numbering "corrected" document:** P.4, L.130

**Reply:** done

**RC1:** P.4, L.148: "industrially most active region of Switzerland" – Insert "the"

**New line numbering "track_changes" document:** P.4, L.157

**New line numbering "corrected" document:** P.4, L.151

**Reply:** done

**RC1:** P.5, L163: "from the latter on a" – Suggest changing to "from this area on a "

**New line numbering "track_changes" document:** P.5, L.171

**New line numbering "corrected" document:** P.5, L165

**Reply:** done

**RC1:** P.7, L.226: "are at a significant distance" – Please remind the reader of distance to the nearest large town e.g. more than 20km.

**New line numbering "track_changes" document:** P.8, L.268

**New line numbering "corrected" document:** P.8, L.255

**Reply:** done. We added: "(…) (i.e. Lucerne as the nearest large town 20 km away) (…)"

**RC1:** P.8, L.255: "were weighed accordingly" – Change to "weighted"

**New line numbering "track_changes" document:** P.9, L.296

**New line numbering "corrected" document:** P.8, L.284

**Reply:** done

**RC1:** P.8, L.255: "to result in a CO emission" – Add the word 'Swiss'

**New line numbering "track_changes" document:** P.9, L.296

**New line numbering "corrected" document:** P.8, L.284

**Reply:** done

**RC1:** P.8, L.264: "in Supplement 4.2." – Better to say Supplement Fig. S2 as 4.2 doesn't exist.

**New line numbering "track_changes" document:** P.9, L.305

**New line numbering "corrected" document:** P.9, L.293

**Reply:** done. We corrected this to: "(…) is given in Supplement 4."

**RC1:** P.8, L.266: "pollution events were summed up" – Please remind the reader that the baseline has been removed to estimate a pollution event.

**New line numbering "track_changes" document:** P.9, L307

**New line numbering "corrected" document:** P.9, L295

**Reply:** done. We corrected the sentence to: "(…) all remaining pollution events above baseline were summed (…)"

**RC1:** P.8, L.280: "(Supplement 4.3)" – Supplement Table S4, 4.3 does not exist

**New line numbering "track_changes" document:** P.10, L.322

**New line numbering "corrected" document:** P.9, L.310

**Reply:** done. We corrected this to: "(…) was greatly reduced (Supplement 4) (…)"

**RC1:** P.8, L.282: Should section 2.4 come before 2.3 as these simulations are used for the residence times?

**New line numbering "track_changes" document:** P.10, L.324/ P.7, L.230

**New line numbering "corrected" document:** P.7, L.220

**Reply:** done. We exchanged section 2.3 and 2.4. We updated the respective cross-references.

**RC1:** P.9, L.304: "Next to total receptor mole fractions" – Not sure I understand the use of the phrase 'Next to', what does it mean?

**New line numbering "track_changes" document:** P.8, L.252

**New line numbering "corrected" document:** P.7, L.242

**Reply:** done. We changed the sentence to: "Besides total receptor concentrations, spatially resolved FLEXPART source sensitivities were used to identify situations in which air masses sampled at Beromünster were dominated by surface contact over the Swiss domain."

**RC1:** P9. L323: "ð• œ'ð• 'œ to the observations" – In the previous paragraph the observations were referred to as 'y', please change for consistency.

**New line numbering "track_changes" document:** P.11, L.367

**New line numbering "corrected" document:** P.10, L.329

**Reply:** done. We changed it to: "and the observations $\chi_o$ in a BI framework"

**RC1:** P9. L325: "model-observation uncertainty" – Adding the word 'respectively' at the end of the sentence will help the reader.

**New line numbering "track_changes" document:** P.11, L.374

**New line numbering "corrected" document:** P.10, L.335

**Reply:** done

**RC1:** P9. L325: I think it would be helpful to the reader to very briefly describe the spatial and temporal covariances used to construct B and R rather than rely on them reading another paper.

**New line numbering "track_changes" document:** P.11, L.375

**New line numbering "corrected" document:** P.10, L.337

**Reply:** Thank you for the remark. We added a paragraph explaining the setup of the covariance matrices.

**RC1:** P10. L333: "(ð• œ' 0 , ð• › ð• ' ¥)" – Here the O is a superscript, previously it was a subscript.

**New line numbering "track_changes" document:** P.12, L.404

**New line numbering "corrected" document:** P.10, L.362

**Reply:** done. We changed the 0 to subscript: "(…) values $(\chi_o,\ Mx)$ (…)"

**RC1:** P10, L349: "countries of the inversion domain" – Consider changing 'of' to 'in'

**New line numbering "track_changes" document:** P.12, L.426

**New line numbering "corrected" document:** P.11, L.383

**Reply:** done

**RC1:** P10. L359: "chi index" – I think this term needs greater explanation with an appropriate reference.

**New line numbering "track_changes" document:** P.13, L.444

**New line numbering "corrected" document:** P.11, L.402

**Reply:** We modified the sentence as follows in order to provide a concise definition. A reference to Berchet et al. (2013), who explored the use of this parameter in inverse modelling of CH4 emissions, was added as well.

The $\chi^2$ index (defined as, $\chi^2 = J(x)\,^2\!/_d$, d being the number of observations) assesses the probability density distribution of the a posteriori model residuals and a posteriori emission differences, which should follow a $\chi^2$ with mean equal to $d/2$.

**RC1:** P10. L367: "different substances was evaluated" – please quantify how this was done.

**New line numbering "track_changes" document:** P.13, L454

**New line numbering "corrected" document:** P.12, L412

**Reply:** This was done semi-objectively, based on the parameters in table S5. The evaluation depends on all these parameters, describing the quality of the inversion; they give us some indication and the final judgement was done based on this. In the text we now especially indicated a threshold of 0.1 for the correlation coefficient ($r^2$), based on which we considered a result reliable or not. Also for the tracer-ratio method we indicated the minimum number of 10 data points incorporated in the calculation for a result to be reliable.

**RC1:** P.12, L.381: "atmospheric concentrations" – consider changing to 'atmospheric mole fraction'

**New line numbering "track_changes" document:** P.16, L. 472

**New line numbering "corrected" document:** P.13, L. 427

**Reply:** For consistency we changed the term "mole fraction" to the term "concentration" throughout the manuscript and the supplement, including the figures.

**RC1:** P.13, L404: "small emissions in Switzerland" – add HFC-152a to this for clarity.

**New line numbering "track_changes" document:** P.17, L.495

**New line numbering "corrected" document:** P.14, L.450

**Reply:** done. We change the sentence to: "This can be explained by small HFC-152a emissions in Switzerland arising only from (…)".

**RC1:** P.13, L413: "there was no notable number of" – consider changing to ' there were no notable'

**New line numbering "track_changes" document:** P.17, L.504

**New line numbering "corrected" document:** P.14, L.459

**Reply:** As we want to refer to the number of the pollution events, we changed the text to: "However, there were only very few pollution events."

**RC1:** P.13, L.415: "emitted as unwanted" – – consider changing to 'emitted as an unwanted'
**New line numbering "track_changes" document:** P.17, L.507
**New line numbering "corrected" document:** P.14, L.461
**Reply:** done

**RC1:** P.13, L.417: "major fraction of the highest events" – please quanitify
**New line numbering "track_changes" document:** P.17, L.508
**New line numbering "corrected" document:** P.14, L.463
**Reply:** done. We wrote: "For $SF_6$, sporadic pollution episodes were observed, with only 17 % of the pollution events greater than 1 ppt, however, showing a high contribution from Switzerland."

**RC1:** P.17, L.499: "mostly used in refrigeration" – 'mostly used as a refrigerant'
**New line numbering "track_changes" document:** P.22, L.594
**New line numbering "corrected" document:** P.18, L.539
**Reply:** done

**RC1:** P.17, L.502: "are invariably higher" – remove the word 'invariant'
**New line numbering "track_changes" document:** P.22, L.598
**New line numbering "corrected" document:** P.18, L.542
**Reply:** done

**RC1:** P.18, L.531: "third highest emissions" – 'third highest Beromünster emission estimate'
**New line numbering "track_changes" document:** P.23, L. 628
**New line numbering "corrected" document:** P.19, L. 569
**Reply:** We adjusted the text accordingly to make it more clear.

**Additional corrections by the authors, apart from the reviewer comments:**
**Abstract:** We improved the abstract text so that it reads better
**Jungfraujoch-based emission estimates:** We updated the reference of (Reimann et al. 2020) to (Reimann et al. 2021) and updated the Jungfraujoch-based emission values to this report, or, where needed, to the newest calculation results for the corresponding years, as these emission values are adjusted and improved constantly for the year before the newest report is published. This is because the Jungfraujoch-based emissions are calculated as a three-year average. The difference to the values listed in this manuscript before adjustment are small, however.
**Beromünster tracer-ratio method:** We used the underlying data-set of carbon monoxide (CO) acquired by the Swiss NABEL network. The instrument is another version of Picarro analyzer, also using cavity ring-down spectroscopy. For the tracer-ratio emission results this makes only a minor difference, but we changed the method description for CO measurements in Section 2.2 "Sampling and Analysis" and updated the new emission results in Table 2.

**Bayesian inversion:** We added more details on the source of the a priori values for specific substances in the text and in the caption of Table 1. The a priori values listed in Table 1 were updated to the originally used UNFCCC and CLIMGAS values, not the a priori values already modified by the inversion calculations, since this might be confusing. The emissions results in Table 2 were updated in the context of the changed modelling.

**Acknowledgements:** We added more details regarding the contributors and funding of this study.

**A few minor corrections**: We corrected single words or punctuation characters throughout the manuscript.

---

## Author Comment (AC2)

**Swiss halocarbon emissions for 2019 to 2020 assessed from regional atmospheric observations**

Dominique Rust[1,2], Ioannis Katharopoulos[1,3], Martin K. Vollmer[1], Stephan Henne[1], Simon O'Doherty[4], Daniel Say[4], Lukas Emmenegger[1], Renato Zenobi[2], Stefan Reimann[1]

[1]Laboratory for Air Pollution/Environmental Technology, Empa, Swiss Federal Laboratories for Materials Science and Technologies, Dübendorf, Switzerland
[2]Department of Chemistry and Applied Biosciences, ETH, Swiss Federal Institute of Technology, Zurich, Switzerland
[3]Institute for Atmospheric and Climate Science, ETH, Swiss Federal Institute of Technology, Zurich, Switzerland
[4]Atmospheric Chemistry Research Group, University of Bristol, Bristol, UK

*Correspondence to*: Stefan Reimann (stefan.reimann@empa.ch)

**Introduction**

Swiss halocarbon emissions for 2019 to 2020 assessed from regional atmospheric observations by Rust et al.

The paper presents the results of an observation/modelling activity aimed at estimating emissions, at the national scale, of a wide range of halocarbons which are ozone-depleting and/or radiatively active gases.

This activity is recognized as relevant from a policy perspective because useful for the validation of national emission inventories and to ascertain the countries' compliance to the international agreements.

The paper is well written and clear, and the research is based on an outstanding and well-established observation activity. However, I have reservations about the methodology used for data analysis and the interpretation of results.

**RC1:** Concerning the methodology, my first question is about the station network. It is well known that reliable regional modelling requires a dense network of stations, which in most regions is not available. As stated in the introduction, in Europe, the AGAGE network continuously measures halocarbons. Beside Jungfraujoch and Mace Head, the network includes two additional sites (Mt Cimone and Ny-Alesund). Moreover, Taunus station in Germany started its monitoring activity in the period covered by the present study. In total, within the study domain, there are three more sites measuring halocarbons in addition to those used by the authors for their analysis.
Considering that the Swiss territory is located between two complex and strong pollutants' source regions, Germany to the north and Italy to the south, and considering that both Mace Head and Tacolneston are far away from sources that might affect the signal at JFJ and BRM and are not able to trace transport from the south, it would be important if the authors could justify the choice of using a less dense network.
**Reply:** The decision of the selection of sites was, on the one hand, based on the quality and completeness of the observations at each site and, on the other hand, on the sensitivity of the sites to Swiss emissions, the main focus of this study. The sites used in this study employ the same Medusa measurement technique for the compounds presented here, whereas two of the other mentioned sites (Taunus and Monte Cimone) rely on alternative measurement techniques that do not result in the same set of reliably observed compounds as presented here. Furthermore, the sites Monte Cimone and especially Ny Alesund are not very sensitive to Swiss emissions. In case of the

former, the Alps form a natural barrier that considerably lowers sensitivities although the site is not very distant from Switzerland itself. It is true that Monte Cimone is sensitive to emissions in northern Italy and that there may be some cross talk from changes in those emissions to emissions in Switzerland, but we considered the much larger sensitivity of the Beromünster observations to outweigh such crosstalk by far. The observations from Ny Alesund are generally not very sensitive to European emissions and the site is generally not used for regional emission estimates in Europe. Concerning the observations from Taunus Observatory, we agree with the reviewer that these are very valuable for the European perspective. In order to analyse their impact on Swiss emission estimates, we ran an additional inversion for HFO-1234yf including continuous observations from Taunus Observatory. Swiss national emissions in this run differed by less than 5 % from the base case. Since HFO-1234yf emissions showed a typical distribution with large emissions in the Benelux area, Germany and Northern Italy, we don't expect the impact of additional observations from Taunus Observatory to have a larger impact for other compounds either.

**RC1:** Have the authors performed comparative tests to determine the sensitivity of the receptor to the source using different sets of stations (including those mentioned above)?

**Reply:** Next to the inversion run including observations from an additional site (Taunus Observatory) we also performed a sensitivity run excluding the sites from the British Isles. For HFO-1234yf the result for Swiss national emissions only differed from the base run by 3 %. However, we saw that including MHD and TAC had a strong impact on a posteriori emissions from the Benelux area and Western Germany. To avoid any kind of cross talk from northwestern Europe to Switzerland it was deemed beneficial to include the two sites in the inversions for all compounds. Further addition of Taunus Observatory had less impact on emissions from northwestern Europe. Hence, for additional reasons given above, Taunus was not included. The results discussed here for HFO-1234yf may not be the representative for all compounds investigated, but performing the same set of sensitivity inversions for all compounds was beyond the scope of the current analysis. In future studies, the selection of observational data should be checked again for specific compounds. We have added the following discussion of observation selection to the manuscript:

"Sensitivity inversions for HFO-1234yf were performed in order to quantify the sensitivity of the a posteriori emissions in Switzerland to the selection of measurement sites. When adding additional observations from the Taunus Observatory in central Germany or when removing the observations from the British Isles, changes in total Swiss emissions were smaller than 5 %. "

**RC1:** Concerning the presentation of results (section 3.2), I do not see any result obtained using the Bayesian inversion at JFJ, neither a comparison between results obtained by Bayesian inversion at BRM and at JFJ. Since the comparison of the results obtained with the two methods (TRM and BI) at BRM highlights relevant differences for many of the compounds considered, with the TRM mostly over-estimating with respect to the BI, it would be useful to ascertain if the same deviations are observed comparing the two methods at JFJ.

**Reply:** The Bayesian inversion combines observations from all mentioned sites to derive spatially resolved emissions. This is in contrast to using information from individual sites in the TRM. We did not perform inversions based on JFJ observations only, given their limited sensitivity to Swiss emissions. However, we agree that the label "BI from BRM" was misleading and have removed it from the manuscript, i.e. we corrected this throughout the manuscript and the supplement.

**RC1:** Comparing the averaged BRM-TRM/BRM-BI fluxes with values obtained at JFJ by TRM only is less meaningful than comparing the average of two methods at BRM with the average of two methods at JFJ.

Therefore, I would like to ask why fluxes evaluated through the Bayesian Inversion at JFJ are not reported in this paper.

**Reply:** As explained above, Bayesian inversions for Swiss emissions were never calculated based on JFJ data only. JFJ observations are not sufficiently sensitive to Swiss emissions alone. This fact was one of the main reasons for the additional measurement campaign at BRM. However, additionally including JFJ (and MHD, TAC) observations in this study helps the inverse modeling to arrive at a more robust solution for Swiss emissions as well.

**RC1:** Concerning the interpretation of the results (page 17), the authors state that there is little consistency between BRM and JFJ data for a subset of three chlorinated gases. However, looking at percentage differences in Table 2, the lack of consistency seems to affect many compounds considered in the study.

**Reply:** On page 23 in the track changes document; Unfortunately this sentence on the lack of consistency towards the Jungfraujoch results was misleading. We meant to say that not only the Bayesian inversion and the Beromünster tracer-ratio results diverge from each other, but that also the Jungfraujoch tracer-ratio results diverge from the latter two. We re-wrote the paragraph to make it clear.

**RC1:** In the conclusions (page 22), the authors state that BRM data provide valuable information for the validation of halocarbon inventories. I agree that the use of a not remote station located in the middle of the Swiss territory at an altitude of 700 m, allows the authors to investigate emissions from the Swiss boundary layer with higher reliability.

However, given the large differences in fluxes evaluated using two different methods, the interpretation of results is quite difficult, and there is a risk that the analysis, rather than supporting the importance of observation-based methods, might lead to some doubts for most of the compounds considered in the study, except for the three most emitted HFCs (134a, 125 and 32) or for the sum of the HFCs.

**Reply:** On page 28 in the track changes document; Thank you for this justified remark. This issue was also extensively addressed in the reply to the second independent reviewer. To meet this point, we adjusted the last sentence of the conclusions and added another subsection (Section 3.2.4 Methods appraisal) to the manuscript discussing the two calculation methods in this light: "Both applied methods, the TRM and the BI, have their advantages and disadvantages. For the TRM we make the assumptions that the analyte and the tracer have similar spatial and temporal emissions sources, and that the transport distance is either sufficiently short for the ratio of analyte and tracer to be preserved until reaching the receptor, or that the transport distance is sufficiently long so that analyte and tracer emissions from multiple sources are well-mixed when reaching the receptor (Sect. 2.4). The Bayesian inversion makes the assumption that emissions are constant in time. For compounds with intermittent emissions, this may lead to reduced model performance. Furthermore, the method seeks to locate emissions in space, guided by a priori information. In the case of large spatial differences between a priori and real emissions, the method will be challenged, once again leading to reduced model performance. We observe, that for most substances, except the major HFCs, the HFOs, and $SF_6$, the TRM result exceeds the BI result. Possible reasons for this are that the assumption of similar emission sources of analyte and CO as the tracer does not hold and/or that the analyte and tracer emissions are not well mixed when reaching the receptor, leading to a distortion of the halocarbon-tracer

ratio. Nonetheless, we used CO as a good universal tracer for many substances. If we used another dispersedly emitted tracer, we would have similar problems. For both calculation methods, we indicated the reliability of the emissions results (Sect. 2.4, 2.5, and Fig. 4) and regarding the most highly emitted substances, we especially consider our results dependable for the major HFCs, the three HFOs, and $SF_6$."

**Specific minor comments**

**RC1:** Pag 8, line 256: the reference Reimann et al., 2020 to be checked or added

**New line numbering "track_changes" document:** P.9, L.297

**New line numbering "corrected" document:** P.8, L.285

**Reply:** We updated it to the publication of Reimann et al. (2021).

**RC1:** Pag 10, line 344: I would use "distributed" rather than calculated

**New line numbering "track_changes" document:** P.12, L.421

**New line numbering "corrected" document:** P.11, L.379

**Reply:** done

**RC1:** Pag 13, line 416: please consider adding the reference Keller et al. (2011) "Evidence for under-reported western European emissions of the potent greenhouse gas HFC-23"

**New line numbering "track_changes" document:** P.17, L.507

**New line numbering "corrected" document:** P.14, L.461

**Reply:** done. We added the Keller et al. (2011) reference.

**Additional corrections by the authors, apart from the reviewer comments:**

**Abstract:** We improved the abstract text so that it reads better

**Jungfraujoch-based emission estimates:** We updated the reference of (Reimann et al. 2020) to (Reimann et al. 2021) and updated the Jungfraujoch-based emission values to this report, or, where needed, to the newest calculation results for the corresponding years, as these emission values are adjusted and improved constantly for the year before the newest report is published. This is because the Jungfraujoch-based emissions are calculated as a three-year average. The difference to the values listed in this manuscript before adjustment are small, however.

**Beromünster tracer-ratio method:** We used the underlying data-set of carbon monoxide (CO) acquired by the Swiss NABEL network. The instrument is another version of Picarro analyzer, also using cavity ring-down spectroscopy. For the tracer-ratio emission results this makes only a minor difference, but we changed the method description for CO measurements in Section 2.2 "Sampling and Analysis" and updated the new emission results in Table 2.

**Bayesian inversion:** We added more details on the source of the a priori values for specific substances in the text and in the caption of Table 1. The a priori values listed in Table 1 were updated to the originally used UNFCCC and CLIMGAS values, not the a priori values already modified by the inversion calculations, since this might be confusing. The emissions results in Table 2 were updated in the context of the changed modelling.

**Acknowledgements:** We added more details regarding the contributors and funding of this study.

**A few minor corrections**: We corrected single words or punctuation characters throughout the manuscript.

---

## Author Response (AR2)

**Swiss halocarbon emissions for 2019 to 2020 assessed from regional atmospheric observations**

Dominique Rust[1,2], Ioannis Katharopoulos[1,3], Martin K. Vollmer[1], Stephan Henne[1], Simon O'Doherty[4], Daniel Say[4], Lukas Emmenegger[1], Renato Zenobi[2], Stefan Reimann[1]

[1]Laboratory for Air Pollution/Environmental Technology, Empa, Swiss Federal Laboratories for Materials Science and Technologies, Dübendorf, Switzerland
[2]Department of Chemistry and Applied Biosciences, ETH, Swiss Federal Institute of Technology, Zurich, Switzerland
[3]Institute for Atmospheric and Climate Science, ETH, Swiss Federal Institute of Technology, Zurich, Switzerland
[4]Atmospheric Chemistry Research Group, University of Bristol, Bristol, UK

*Correspondence to*: Stefan Reimann (stefan.reimann@empa.ch)

**Reply to editor:**

Dear Mr. Grooß,

thank you very much for your clarifying remarks. Below please find our answers.

**(1) Would you judge the average emissions in Table 2 as your recommended best values taking in account advantages and disadvantages of the two methods?**

Yes, as written in Sect. 3.2.4, both top-down methods incorporate specific disadvantages and thus uncertainties. Since it is very difficult to quantify these method uncertainties arising from made assumptions, we stated the average of the results calculated with both independent methods as our best emission values, thereby also increasing the uncertainty to each average value listed in Table 2. In addition, we indicated where the uncertainty range of the average value does not overlap with the individual results, which is in many cases when we also pointed out reduced reliability of our calculated results, evaluated based on the parameters described in the respective sections. We added the following text to Sect. 3.2.4 of the manuscript: "With both independent top-down approaches incorporating method uncertainties due to the made assumptions, we stated the average of the individually calculated TRM and BI results as our best emission value (Table 2), thereby also increasing the uncertainty to each emission value. In addition, we indicated where the uncertainty range of the average emission value does not overlap with the individual results, which often is the case when we pointed out reduced reliability of our calculated results."

**(2) It may be valuable to provide the results, especially the emission maps of the BI method also in the form of data files in addition to the used data in the observation data. This may help in constructing updates of the emission inventories.**

We added additional data results for the Bayesian inversion to the Zenodo repository. Therefore a new Zenodo version (2.0.0) had to be created with a new DOI (https://doi.org/10.5281/zenodo.5843548), which was also referenced accordingly in the data availability section of the manuscript. The Zenodo repository now contains the Beromünster measurement data file and information for the Bayesian inversion. The data files for the Bayesian inversion are provided in netCDF format for the 28 individual substances discussed in the paper. Each file contains the a priori and a posteriori emissions as used or calculated in the Bayesian inversion. Data are provided on the grid used in the inversion (irregular longitude/latitude). Metadata are included as netCDF attributes. The netCDF files follow the CF conventions and should be readable with any netcdf interface/tool.